# Development of Indian summer monsoon precipitation biases in two seasonal forecasting systems and their response to large-scale drivers

Richard J. Keane[1,2], Ankur Srivastava[3], Gill M. Martin[1]

[1]Met Office, Exeter, UK
[2]School of Earth and Environment, University of Leeds, UK
[3]Indian Institute of Tropical Meteorology, Ministry of Earth Sciences, Pune, India

*Correspondence to*: Richard J Keane (richard.keane@metoffice.gov.uk)

**Abstract.** The Met Office Global Coupled Model (GC) and the NCEP Climate Forecast System (CFSv2) are both widely used for predicting and simulating the Indian summer monsoon (ISM), and previous studies have demonstrated similarities in the
biases in both systems at a range of time scales from weather forecasting to climate simulation. In this study, ISM biases are studied in seasonal forecasting setups of the two systems, in order to provide insight into how they develop across time scales. Similarities are found in the development of the biases between the two systems, with an initial reduction in precipitation followed by a recovery associated with an increasingly cyclonic wind field to the north-east of India. However, this occurs on longer time scales in CFSv2, with a much stronger recovery followed by a second reduction associated with sea surface
temperature (SST) biases, so that the bias at longer lead times is of a similar magnitude to that in GC. In GC, the precipitation bias is almost fully developed within a lead time of just eight days, suggesting that carrying out simulations with short time integrations may be sufficient for obtaining substantial insight into the biases in much longer simulations. The relationship between the precipitation and SST biases in GC seems to be more complex than in CFSv2, and is different during the early part of the monsoon season from during the later part of the monsoon season.

The relationship of the bias with large-scale drivers is also investigated, using the Boreal Summer IntraSeasonal Oscillation (BSISO) index as a measure of whether the large-scale dynamics favours increasing, active, decreasing or break monsoon conditions. Both models simulate decreasing conditions the best and increasing conditions the worst, in agreement with previous studies and extending these previous results to include CFSv2 and multiple BSISO cycles.

### Copyright statement

# 1 Introduction

The Indian summer monsoon (ISM) is one of the most challenging meteorological phenomena to simulate, with many current general circulation models (GCMs) having substantial systematic biases (Jain et al., 2019; Katzenberger et al., 2021; Mitra, 2021; Watterson et al., 2021; Choudhury et al, 2022). Two examples of GCMs with a persistent low-precipitation bias for the ISM are the Met Office Global Coupled Model (GC, Williams et al., 2015; 2017) and the NCEP Climate Forecast System (CFSv2, Saha et al., 2014). This has been shown to occur in these models across a range of time scales—with common

accompanying features including an anti-cyclonic bias and a high-precipitation bias over the ocean to the south of India—from weather forecasts (Kar et al., 2019; Keane et al., 2019; 2021; Abhilash et al., 2014) to seasonal and climate simulations (Walters et al., 2019; Martin et al., 2021; Swapna et al., 2018; Sahana et al, 2019). The purpose of the present study is to investigate this low-precipitation bias in detail in seasonal simulations, to provide insight into how it develops from shorter to longer time scales.

Many previous studies have investigated the skill of seasonal forecasts, using GC or CFSv2, in predicting the ISM, and these studies have generally demonstrated similar biases to those in weather and climate simulations (Abhilash et al., 2014; George et al., 2016; Ramu et al., 2016; Johnson et al., 2017; Srivastava et al., 2017; Jain et al., 2019; Chevuturi et al., 2019; Martin et al., 2021; Joseph et al., 2023; Kolusu et al., 2023). Despite these biases, the seasonal forecasts do show skill, particularly at shorter lead times of up to two weeks (George et al., 2016; Rao et al., 2019; Joseph et al., 2023; Kolusu et al., 2023), and can

reasonably well simulate the northward propagation of the monsoon intraseasonal oscillation (Abhilash et al., 2014; Sabeerali et al., 2013; Srivastava et al., 2023), low-pressure systems (Srivastava et al., 2017; 2023) and the monsoon onset (Menon et al., 2018; Chevuturi et al., 2019; Pradhan et al. 2017). George et al. (2016) attributed this skill in CFSv2 to correctly capturing connections with the El Nino Southern Oscillation, with Indian Ocean coupled dynamics not adequately represented in CFSv2, and similar behaviour was demonstrated for GC by Johnson et al. (2017).

The atmospheric biases have been shown to be associated with cold sea surface temperature (SST) biases over the Indian Ocean in CFSv2 (George et al., 2016; Srivastava et al., 2017) and GC has also been shown to develop SST biases in seasonal forecasts within the first 30 days (Martin et al., 2021); Johnson et al. (2017) attributed incorrect SST anomalies to a lack of wind forcing on the SSTs. We investigate this interdependence of atmospheric and oceanic biases in the seasonal forecasting systems here by carrying out a systematic investigation of how precipitation, wind and SST biases vary with forecast lead time.

Martin et al. (2021) investigated systematic biases for the Asian summer monsoon, in GC configurations on a range of time scales. They showed that, while the biases in seasonal forecasts and climate simulations generally have similar patterns and magnitudes, those over India have larger magnitudes in the climate simulations, indicating that these biases could have a substantial dependence on how far ahead of the monsoon season the simulation is initialised. Meanwhile, Chattopadhyay et al. (2016) demonstrated an intriguing finding that CFSv2 produces better ISM forecasts at longer lead times than at shorter lead

times. It is therefore important to look at the lead-time development of the seasonal forecast biases in detail to investigate how they are affected by errors in different physical processes occurring on different time scales.

A more direct aim of this study is to follow Keane et al. (2021), who showed that the reduction in mean ISM precipitation with forecast lead time in weather forecasts using the atmosphere and land components of GC has a strong and coherent dependence on the phase of the Boreal Summer Intraseasonal Oscillation (BSISO; Wang & Xie, 1997; Kikuchi et al., 2012; Lee et al.,

2013; Kikuchi, 2020; Kikuchi, 2021). They showed that the precipitation is initially too high for all phases, and the subsequent reduction is strongest for phases 2–4, corresponding to broadly increasing monsoon activity, so that by the end of the forecast there is a substantial low-precipitation bias. For phases 5–7, corresponding to broadly decreasing monsoon activity, the reduction is much weaker so that the value at the end of the forecast is actually quite close to observed values. It is therefore of interest to investigate longer forecasts with the same model to investigate how this behaviour develops beyond the 7 days

of the weather forecasts, as well as the influence of ocean-atmosphere coupling. This study provides an opportunity to do this, as well as to determine whether a similar dependence of the precipitation bias with BSISO phase occurs in CFSv2.

The manuscript proceeds by describing the data evaluated in this study and the methods used to perform the evaluation. The results of the evaluation are then presented, followed by a discussion and concluding remarks.

## 2 Data

The GC setup for producing seasonal forecasts is GloSea, described in detail by MacLachlan et al. (2015). The forecasts are calibrated by running a set of hindcasts initialised at the same time of year as the forecast for a range of previous years using the same model version, so that the model version's statistical errors relative to the relevant seasonal climatology can be determined. Operationally, the hindcasts are initialised as a 7-member ensemble on the 1st, 9th, 17th and 25th of each month for the year range 1993–2016. Hindcasts are used in this study, rather than forecasts, as they provide a large data set and we are

interested in the development of biases in the dynamical model itself rather than the quality of the post-processed forecasts. Operational upgrades to GloSea have always been carried out less often than once per year, meaning that there is at least one full year's worth of hindcasts available for each version. In this study we assess two versions, GloSea5 based on GC2.0 (Williams et al., 2015), using hindcasts generated during the period from November 2017 to August 2018, and GloSea6 based on GC3.2 (closely related to GC3.0 and GC3.1, described by Williams et al., 2017), using hindcasts generated during the

period from November 2021 to August 2022. The atmospheric model resolution of both GloSea versions is N216, corresponding to grid spacings of approximately 70 km in the tropics. ERA-interim reanalysis fields are used to provide the (unperturbed) initial conditions and perturbations between the members are produced through the use of a stochastic kinetic energy backscatter scheme (Bowler et al., 2009). An upgrade to the soil moisture initialisation (described by Gautam et al., 2023) was implemented in 2019 so GloSea6 includes this upgrade whereas the version of GloSea5 evaluated here does not.

Hindcasts have also been produced using CFSv2. The atmospheric model spectral resolution is T382, corresponding to grid spacings of approximately 38 km in the tropics. Two hindcasts were initialised, at 00UTC and 12UTC, roughly every 5 days from February to August for the year range 2002–2015; the precise initialisation dates are provided in Table A1. This means that there are roughly half as many hindcasts initialised during the period February–August each year for CFSv2 (74) as for

GloSea (196), but we compare the two systems directly in the main manuscript and show evidence in Appendix A that using only three of the seven GloSea ensemble members (so giving 84 hindcasts initialised during February–August) does not affect the key conclusions of the paper (Figs. A4, A12). We restrict our evaluation to the period 2002–2015 but, again, show evidence in Appendix A that these years are representative of the full period for which GloSea hindcasts are available (Figs. A5, A13). In this study we evaluate the behaviour of the hindcasts as a function of lead time for given valid time periods. To do this, for each lead time we average the hindcast values over all initialisation dates with a hindcast of the corresponding length occurring during the relevant valid time period. We focus on valid times during June–August, and the total hindcast length is 216 days for GloSea and 208 days for CFSv2. This means that the earliest GloSea hindcasts used are initialised on 1$^{st}$ November the previous year, where the longest lead times extend just into the beginning of June, and the number of available hindcasts for every lead time is approximately the same (12 or 13). However, since the CFSv2 hindcasts are only available from February, hindcasts at the longest lead times are only available later in the June–August period (for example, lead times longer than 150 days are only available for July and August valid times).

We also carry out some evaluation of the hindcasts by taking all those initialised during a certain period and categorising them according to the BSISO state at the start of the hindcast. We look at how the precipitation varies as the hindcast develops, up to a hindcast time of 60 days, averaged over all hindcasts within each category, and compare with the observed precipitation averaged over the corresponding dates.

The purpose of this study is to investigate how model quantities change as the hindcast develops (whether looking at a given set of valid times and increasing the lead time by looking at different hindcasts, or looking at a given set of initialisation times and increasing the integration times of the same hindcasts), rather than to evaluate their performance with respect to observations, which is generally already well known at least in the broadest sense. However, we do compare with observed precipitation, using the IMERG data set (Huffman et al., 2019), in order to provide a baseline for the modelled quantities.

In this study, observed BSISO data are taken from https://iprc.soest.hawaii.edu/users/kazuyosh/ISO_index/data/BSISO_25-90bpfil_pc.extension.txt. The BSISO index is calculated using temporally bandpass filtered observed outgoing longwave radiation (OLR). An extended empirical orthogonal function (EEOF) analysis is carried out on the data for June–October, and, for any given day, principal components can be obtained by projecting the OLR field onto each EEOF. The first two principal components are normalised and can then be plotted on orthogonal axes. A phase and amplitude can then be defined as the azimuthal and radial coordinate of the plotted point on the orthogonal axes. The full method is described by Kikuchi et al. (2012) and Kikuchi (2020).

In practice, the azimuthal coordinate is divided into eight discrete phases, with phases 4 & 5 generally corresponding to increased average precipitation over India and phases 8 & 1 corresponding to reduced average precipitation over India (Kikuchi et al., 2012). We therefore use the phases as an indicator of whether the large-scale dynamics favours enhanced, increasing, suppressed or decreasing convection over India and categorise the hindcasts using two different methods, one using the phase at the hindcast valid time and one using the phase at the hindcast initialisation time, as described above.

## 3 Results

### 3.1 Overall behaviour

The overall biases are shown in Fig. 1 (upper panels), for initialisations during May, and in Figs. A1–A3 for different
initialisation months. There is a low-precipitation bias over most of India and the northern Bay of Bengal, which intensifies
somewhat with increasing lead time (i.e., earlier initialisation time) and is less bad in GloSea6 than in GloSea5. The spatial
pattern of the precipitation bias is broadly similar for the different initialisation times. All models have a high-precipitation
bias over the ocean to the south of India and, to some extent, over the south-western coast of India. The wind biases are shown
against ERA5 winds (Hersbach et al., 2023): there is some sign of an anticyclonic bias over India in all three datasets.

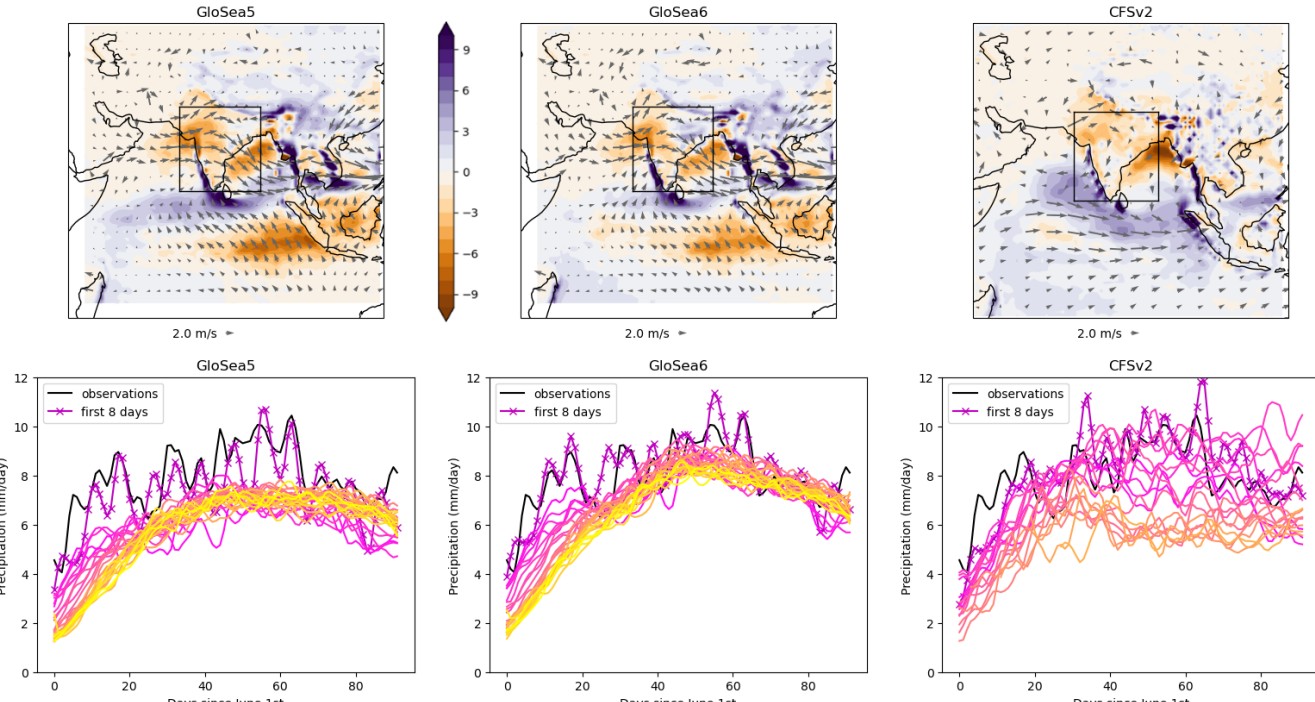

**Figure 1. Top row: Overall biases averaged over valid times in June–August 2002–2015 and initial times during May. Variables are
precipitation in mm/day (bias against IMERG observations) and 850-hPa horizontal wind (bias against ERA5 reanalysis). Bottom
row: Seasonal cycle of precipitation in hindcasts and IMERG observations, averaged over June–August 2002–2015, 8N–29N, 69E–
89E (region denoted by the black boxes in the panels in the top row). Hindcasts are grouped into 8-day lead time sections and
depicted in colours ranging with increasing lead time from magenta (darker, for lead times of 1–8 days) to yellow (lighter, for lead
times of 199–216 days). Quantities are smoothed in the time-of-year direction with a 1-2-1 filter.**

The precipitation averaged over latitudes 8N–29N and longitudes 69E–89E is plotted as a function of time of year (within
June–August), for the three models, in Fig. 1 (lower panels). There is a clear reduction in precipitation after the first 8 days in
GloSea, and this is largest earlier in the monsoon season. In CFSv2 there is also a reduction, which occurs more gradually with
increasing lead time. It is also striking how much less variable the precipitation is at longer lead times, particularly in GloSea.

This may reflect the reduced signal-to-noise ratio at longer lead times, such that averaging over all years and all ensemble members results in a smoother time variability.

The precipitation averaged over latitudes 8N–29N and longitudes 69E–89E, and over valid times during June–August 2002–2015, is shown for each hindcast lead time in Fig. 2. This is the region investigated in detail by Keane et al. (2021) and is used in this study when looking at spatially averaged quantities. Versions of Fig. 2 using fewer GloSea members and using a longer
range of GloSea years are shown in Appendix A (Figs. A4 and A5, respectively), and are very similar. Because hindcasts are only initialised on certain dates within each system, hindcasts at a given lead time will only be available on a subset of the 92 days during June–August each year. For example, in GloSea, at 10 days' lead time there are hindcasts available on 4th, 11th, 19th, 27th June, 5th, 11th, 19th, 27th July, and 4th, 11th, 19th, 27th August; at most lead times there are 12 hindcasts available, with 13 available at some lead times due to the slight irregularity in the rate of initialisation of the hindcasts. The rate of initialisation
for the CFSv2 hindcasts is slightly more irregular, so that there are 15–18 dates available for each lead time up to 137 days (after which the number of available dates gradually decreases, as described in the previous section). The dotted lines therefore show the observed precipitation on the same subsets of dates, and provide an estimate of how much this variation in available dates with lead time should affect the hindcast.

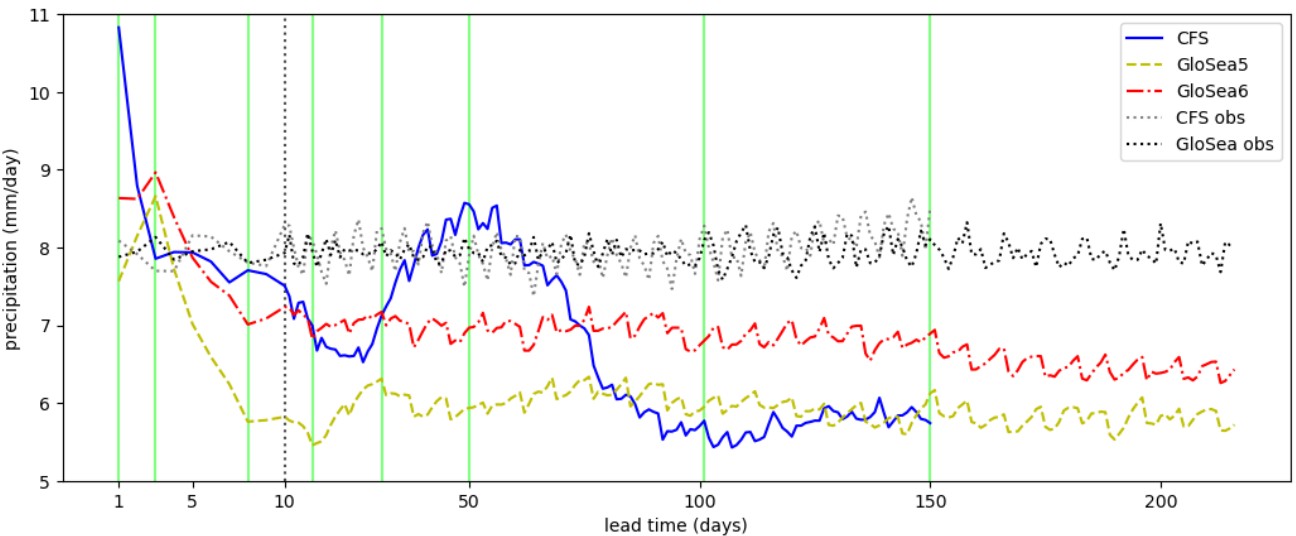

**Figure 2. Variation of precipitation (averaged over 8N–29N and 69E–89E and June–August 2012–2015) with hindcast lead time in different seasonal forecasting systems. The dotted lines show observed (IMERG) values averaged over the dates on which hindcasts were available in the specified system at that lead time. The green vertical lines show the lead times used in Figs. 3–6 (1, 3, 8, 16, 31, 50, 101 and 150 days). Note that the horizontal scale is larger for lead times less than 10 days than for lead times greater than 10 days (so that it is different either side of the vertical dotted grey line).**

Looking at Fig. 2, CFSv2 and GloSea5 reach a similar low-precipitation bias by the end of the hindcast, but this takes longer to develop in CFSv2. GloSea6 also has a low-precipitation bias but this is much improved compared with GloSea5. Both GloSea systems have increasing precipitation at very short lead times and then all systems have sharply decreasing precipitation during the first 8 days or so. The development is then characterised by a slower decrease followed by a recovery, but this is

much stronger, and takes longer to develop, in CFSv2 and is very weak in GloSea6. In CFSv2 there is then a decrease followed by fairly constant values from about 100 days, while in GloSea the precipitation is fairly constant from about 30 days.

### 3.1.1 Changes during specific stages of the hindcast

In order to investigate the behaviour in more detail, the hindcasts are divided into different stages, following the evolution shown in Fig. 2. The first two days are characterised by increasing precipitation in GloSea and decreasing precipitation in CFSv2. In GloSea this may be due to a 'spin-up' from the reanalysis initial state toward behaviour more representative of GC. Days 3–8 account for most of the low-precipitation bias in both GloSea setups. Days 8–16 have fairly constant precipitation in GloSea and decreasing precipitation in CFSv2, while days 16–31 have a small increase in precipitation in GloSea and fairly constant values in CFSv2. The subsequent period of days 31–50 has increasing precipitation in CFSv2, then there is a period of approximately fifty days (50–101) of decreasing precipitation in CFSv2. These two periods have fairly constant precipitation in GloSea and the period from 100 days onwards has fairly constant precipitation in all setups.

To investigate this further, maps of precipitation, 850-hPa wind and SST, in the form of differences between pairs of lead times corresponding to the green vertical lines in Fig. 5, are plotted in Figs. 3–6. For the final two pairs (Days 50—101 and Days 101—150) differences are shown between 31-day averages, centred respectively on the initial and final time, since at such long lead times individual days are less relevant than longer time averages. The absolute values are plotted in Figs. A6–A11 and are characterised by a prevailing westerly flow over India and widespread precipitation, with the largest amounts on the west coast and north-east of India and the Bay of Bengal (BoB).

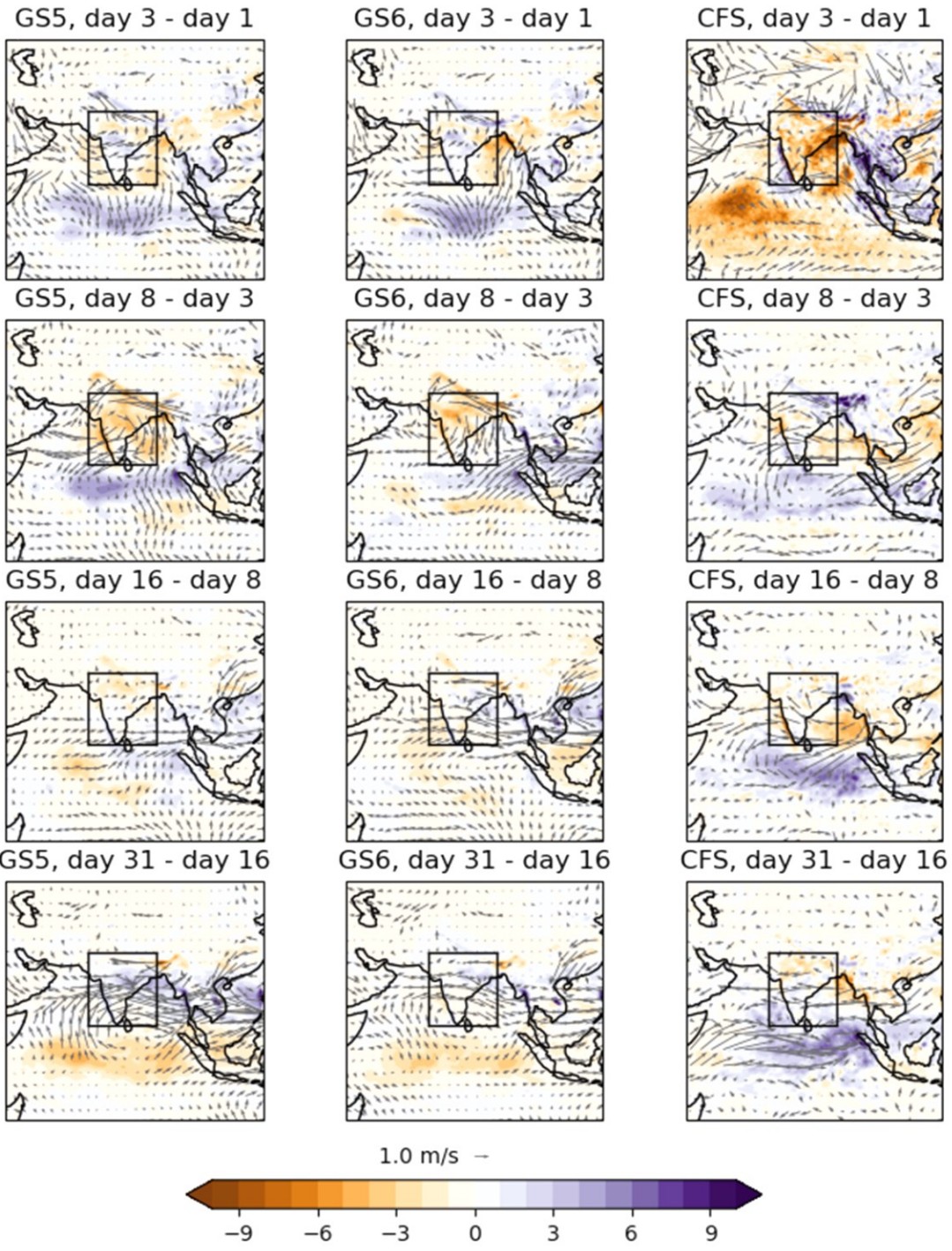

**Figure 3. Precipitation differences (colours, in mm/day) and 850-hPa wind differences (vectors) in GloSea5 (left column), GloSea6 (centre column) and CFSv2 (right column), between the hindcast lead times shown, averaged over June–August 2002–2015. The black boxes show the evaluation region (8N–29N, 69E–89E) used throughout this study.**

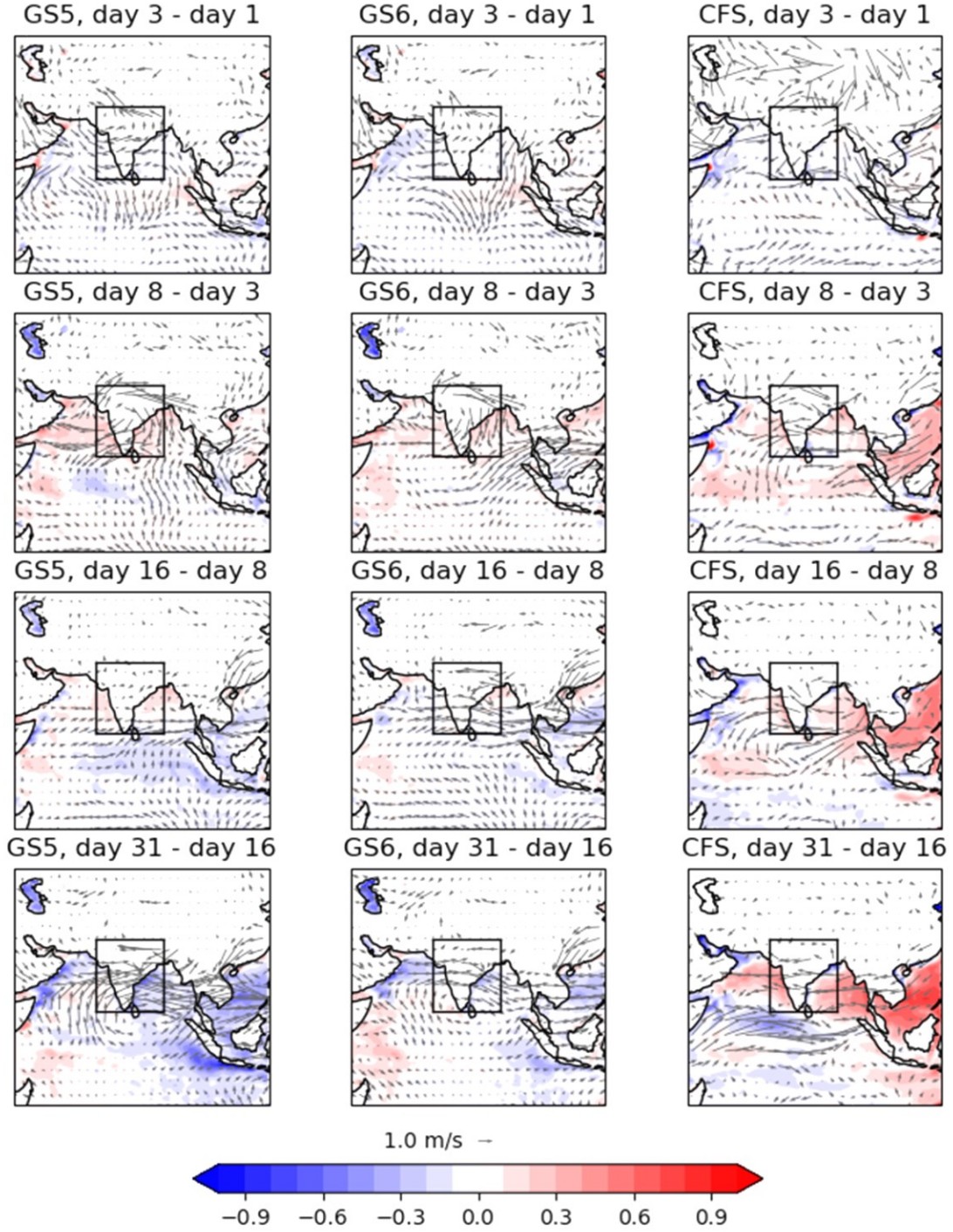

190

**Figure 4. SST differences (colours, in K) and 850-hPa wind differences (vectors; note that this field is identical to that in Fig. 3) in GloSea5 (left column), GloSea6 (centre column) and CFSv2 (right column), between the hindcast lead times shown, averaged over June–August 2002–2015. The black boxes show the evaluation region (8N–29N, 69E–89E) used throughout this study.**

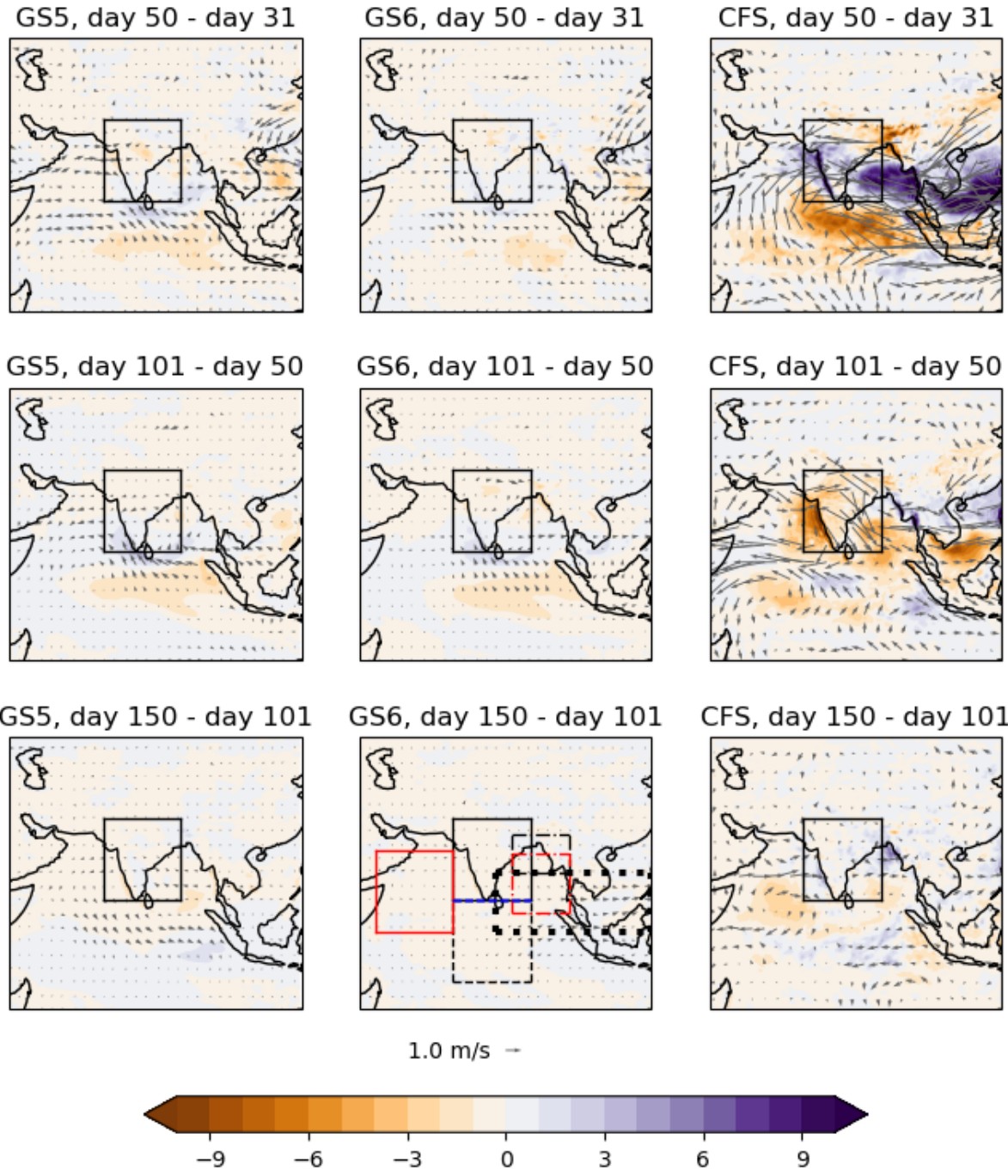

**Figure 5. Precipitation differences (colours, in mm/day) and 850-hPa wind differences (vectors) in GloSea5 (left column), GloSea6 (centre column) and CFSv2 (right column), between two fields at (top row) and averaged over 30 days centred on (centre and bottom rows) the two hindcast lead times shown above the panel. The black boxes show the evaluation region (8N–29N, 69E–89E) used throughout this study and the boxes in the bottom centre panel show the regions used in Figs. 7, 8, 10, 11 & 12.**

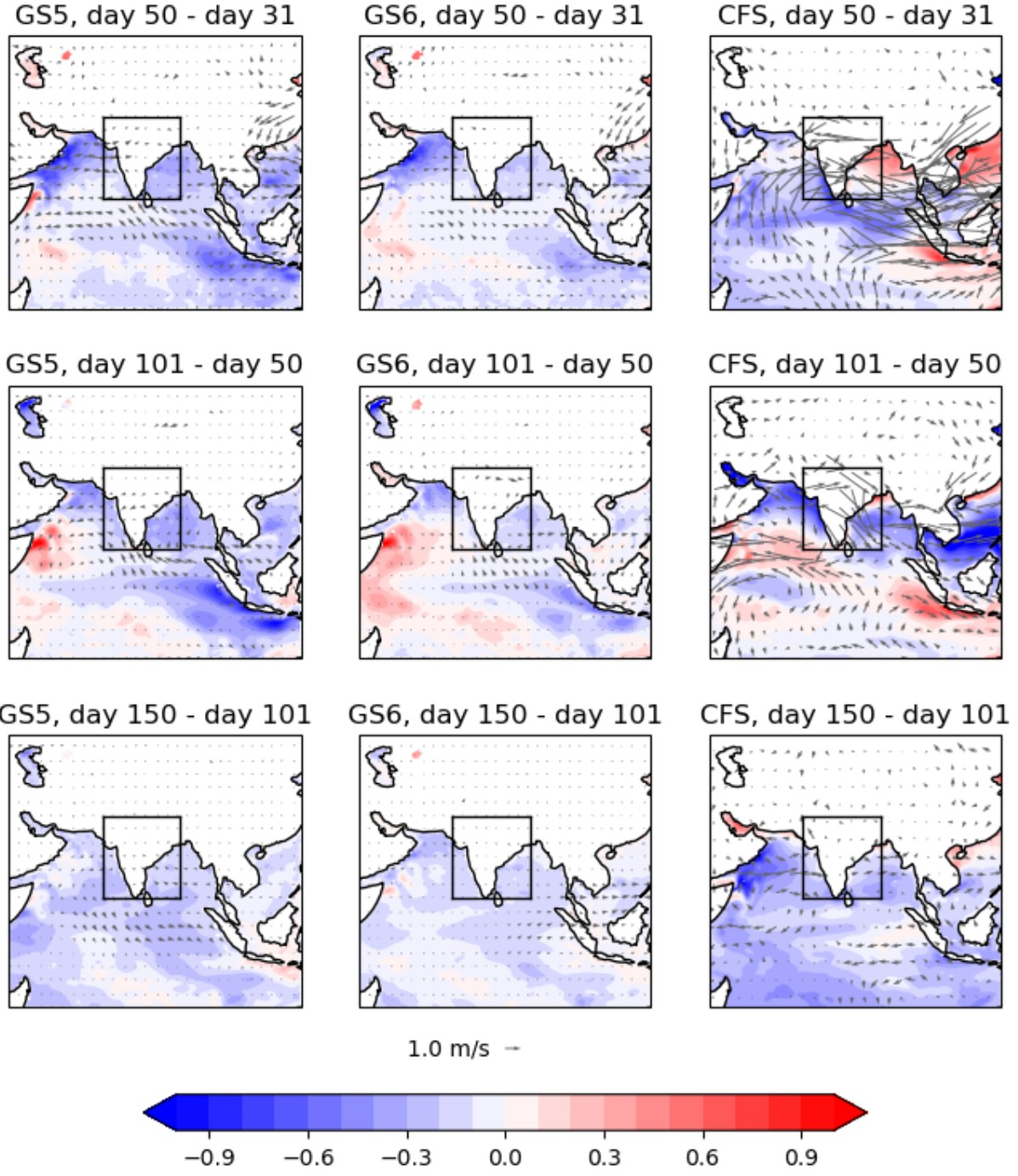

**Figure 6. SST differences (colours, in K) and 850-hPa wind differences (vectors; note that this field is identical to that in Fig. 5) in in GloSea5 (left column), GloSea6 (centre column) and CFSv2 (right column), between two fields at (top row) and averaged over 30 days centred on (centre and bottom rows) the two hindcast lead times shown above the panel. The black boxes show the evaluation region (8N–29N, 69E–89E) used throughout this study.**

During days 1 to 3, the overall westerly flow strengthens in all three systems (albeit rather less coherently in CFSv2), but diverges away from India, in that it is too southerly in the north and too northerly in the south. In GloSea this is accompanied by an increase in precipitation over India and the Equatorial Indian Ocean (EIO), whereas in CFSv2 there is a sharp decrease. There is also increased convergence into the EIO in GloSea. This time period is generally too short for any SST changes to develop.

During days 3 to 8, most of the precipitation reduction occurs in GloSea and it is accompanied by an anticyclonic bias, as seen in previous studies of GC) on weather and climate time scales. The precipitation bias is considerably reduced in GloSea6 but the anticyclonic bias is similar to that in GloSea5. The behaviour is similar in CFSv2 but with rather less coherent patterns. All systems show a decrease in the westerly flow into the south of India, although this is less pronounced in GloSea6. The behaviour during the first 8 days in GloSea is consistent with that in the atmosphere-only weather forecasts studied by Keane et al. (2019; 2021).

Days 8 to 16 represent a relatively quiet period for GloSea in terms of precipitation, but v5 has an increase in anticyclonic flow whereas v6 becomes more cyclonic and the westerly flow into India increases, mitigating the bias in the previous period. CFSv2 also has small precipitation changes over India but there is a decrease over the BoB and an increase over EIO. The flow also becomes more anticyclonic to the east of India. The reduction in precipitation over the BoB is accompanied by an increase in SST and a reduction in the flow from India. SST changes are still very small in GloSea, but with some notable increase over the EIO.

Days 16 to 31 show an increase in the westerly flow in GloSea, bringing increased precipitation over the western coast, and an increase in cyclonic flow, bringing increased precipitation to the very east of India and northern BoB. These features are more pronounced in v5, which could be related to the fact that the reduction in precipitation during earlier periods is stronger, and so there is more scope for a recovery in precipitation. CFSv2 looks very different, with decreased westerly flow over India and reduced precipitation over the northern BoB. The flow to the south of India is more westerly, with (as in the previous period) an increase in precipitation. A similar pattern was seen in the overall bias in CFSv2 simulations carried out by Hari Prasad et al. (2021). SST changes are also different, with increases over the Arabian Sea and BoB in CFSv2 (and decreases further south and along the eastern coast of India) and decreases everywhere in GloSea.

Days 31 to 50 show very little change in GloSea, whereas there is much more change in CFSv2, which has an increase in westerly flow over India and an increase in cyclonic flow over eastern India and the BoB. This is similar to what is seen in GloSea over the previous two periods, but positioned further south and accompanied by an even stronger increase in precipitation. The precipitation decreases over the EIO so that overall the change is in the opposite direction to (and stronger than) that in the previous period in CFSv2. The SST and precipitation changes match very closely in CFSv2, except over the very north of the BoB and Bangladesh and eastern India, and along the western coast of India.

Days 50 to 101 also show little change in GloSea, with small decreases in precipitation in northern India and small increases in southern India. The flow into the southern part of the western side of the box changes direction somewhat so that air is advected more into the western part of the southern side of the box. There is much more change in CFSv2, with a substantial

reduction in precipitation; the pattern looks similar to that in GloSea during days 3 to 8, but with the anticyclonic flow difference and the strongest precipitation reduction centred further west, over the Arabian Sea. The SST and precipitation differences again look remarkably similar in CFSv2.

From day 101 onwards, both systems have roughly constant precipitation within the analysis region used in Fig. 2 but, whereas in GloSea there is very little change in the wind or precipitation fields, in CFSv2 there are changes in both wind and precipitation fields, with the precipitation changes cancelling each other when averaged over the region as a whole. In CFSv2 there is a reduction in westerly flow over India, which seems to cause a reduction in precipitation over the western coast, but an increase over the peninsula, so it could be that less moist air is advected out to the east. Both systems show widespread decreases in SST.

### 3.1.2 Physical interpretation

To investigate the interplay between SSTs, precipitation and wind further, quantities averaged over selected regions are plotted as a function of lead time in Fig. 7. It can be seen that the initial decrease in precipitation over India is mirrored by an increase in precipitation over the EIO in all models (with both occurring over a longer time scale in CFSv2). There is also increasing flow from India to the EIO, suggesting that the increased precipitation over the EIO draws moisture away from India and could be strongly associated with the low-precipitation bias over India. This agrees with previous work (Bollasina & Ming, 2012; Bush et al., 2015; Martin & Rodriguez, 2024) and suggests that this phenomenon occurs very early in any model simulation. In all models, changes in the westerly flow over the Arabian Sea are followed closely by the precipitation both over India as a whole and over the north-east India (NEI) region that seems to be largely responsible for the recovery in precipitation. This suggests that its initial decrease may contribute to the low-precipitation bias, while its subsequent increase could be contributing strongly to the later recovery in precipitation.

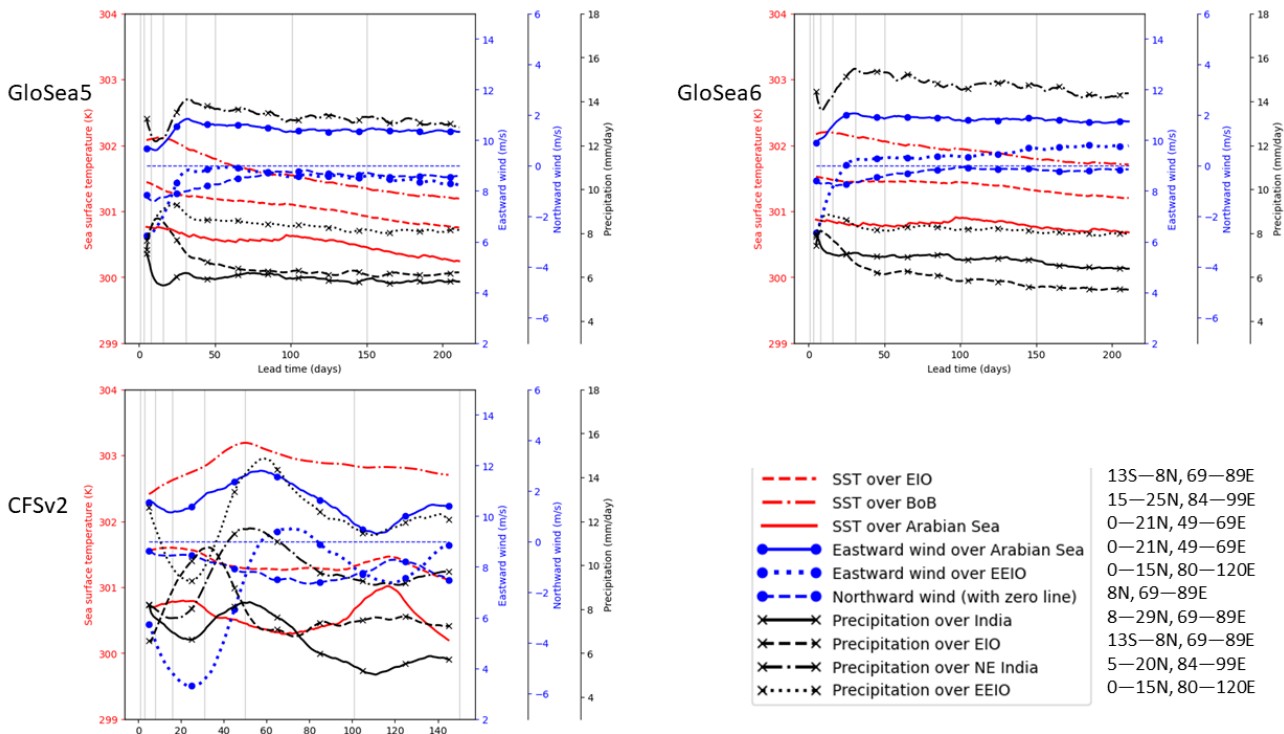

**Figure 7. SST, 850-hPa wind and precipitation averaged over different regions (shown in Fig. 5), averaged over June–August 2002–2015, as a function of lead time. A smoothing is applied by taking a 9-day running average in the lead-time direction.**

The flow out of India into the EIO reduces after about 8 days in GloSea, and is partly connected to the reduction in precipitation over the EIO. This reduction in precipitation is associated with a decrease in SST over the same region through a compensating coupled feedback. There is some evidence that decreasing SSTs over the BoB are similarly related to decreasing precipitation over NEI.

The improved performance in GloSea6 over GloSea5 (smaller precipitation reduction over India) is accompanied by a smaller increase in precipitation over the EIO (and a weaker flow from India out to the EIO) and a smaller decrease in westerly wind over the Arabian Sea, providing further evidence that these are important factors in the low-precipitation bias. There is also less of a reduction in SSTs in all the locations analysed, consistent with the smaller changes in precipitation.

Napusetty et al. (2016) used low-resolution CFSv2 seasonal hindcasts and investigated the lead-time dependence of the moisture transport bias in the Tropical Indian Ocean. They found that the positive bias in moisture transport in the EIO increases systematically from June to August. This bias causes reduced moisture availability to the weakened cross-equatorial monsoon flow, and the strong dry bias in rainfall over the Indian landmass can be partly attributed to this strong positive bias in moisture transport. Interestingly, the bias in moisture transport over EIO is at its maximum in July for May initialization and August for June initialization, compared to other initialization months, which roughly corresponds to a lead time of ~30–90 days. Similar to their findings, the high-resolution version of CFSv2 used here also exhibits a strong bias in vertically integrated moisture transport over EIO (figure not shown). We explore the flow into a box over the eastern EIO (0-15N, 80–

120E, denoted EEIO) and contrast it with the strength of the Findlater jet (49E–69E, 0N–21N, the "Arabian Sea" box). Generally, during JJA, the precipitation over India closely follows the influx of moisture from the Arabian Sea, such that the precipitation curve and the U850 and V850 (over the Arabian Sea, V850 not shown here) curves are in sync. However, the peak in the winds over the Arabian Sea occurs a few days later compared to the peak in precipitation over India, the reasons for which are not clear. The recovery in precipitation in CFSv2 during days 31–50 and the subsequent decline are associated with the strength of the Findlater jet and the associated moisture influx. The recovery in precipitation over EIO (~30–51 days lead time) and the subsequent decline is related to the large influx of moisture over this region by the zonal winds. The decline is also associated with increased flow eastward into the eastern EIO; contrastingly, this flow does not change substantially in GloSea, particularly after 30 days lead time.

Figure 8 shows scatter plots of 850-hPa eastward wind over the Arabian Sea against precipitation over India, coloured by SST over the Arabian Sea. These show individual hindcasts over the full range of years and dates available, for different lead time ranges, in order to investigate how the relationship between the three variables develops within the models as the hindcasts progress. A plot for observed/reanalysed values is included for comparison: this uses IMERG for precipitation (Huffman et al., 2019), ERA5 for 850-hPa winds (Hersbach et al., 2023) and a satellite-derived dataset for SST (Copernicus, 2019).

There is a clear correlation between wind and precipitation in all models at the earliest lead times, agreeing with what is seen in the observations. This correlation weakens, however, as the hindcasts develop, with the weakening being greater in GloSea5 than GloSea6. This suggests that the recovery in westerly wind over the Arabian Sea, seen in Fig. 10, could lead to a stronger recovery in precipitation if the relationship between the two variables were simulated correctly in the models.

The SST ranges vary between the models and between models and observations, but there is a general tendency for higher temperatures to be associated with weaker winds and less precipitation. There is also some evidence from Fig. 8 that the relationship between SST and both wind and precipitation is too strong, as the temperature values vary more coherently with wind and precipitation, in the models than in the observations, particularly at longer lead times. In CFSv2 this is particularly the case for lead times of 51 to 101 days, when the precipitation decreases strongly with lead time, and the temperature range here is also largest. This provides further evidence that SSTs are an important factor in this secondary decrease in precipitation, and suggests that the coupling between the ocean and atmosphere components could be too strong (for example, the effect of stronger winds leading to evaporative cooling could be overestimated) in this region in all the models.

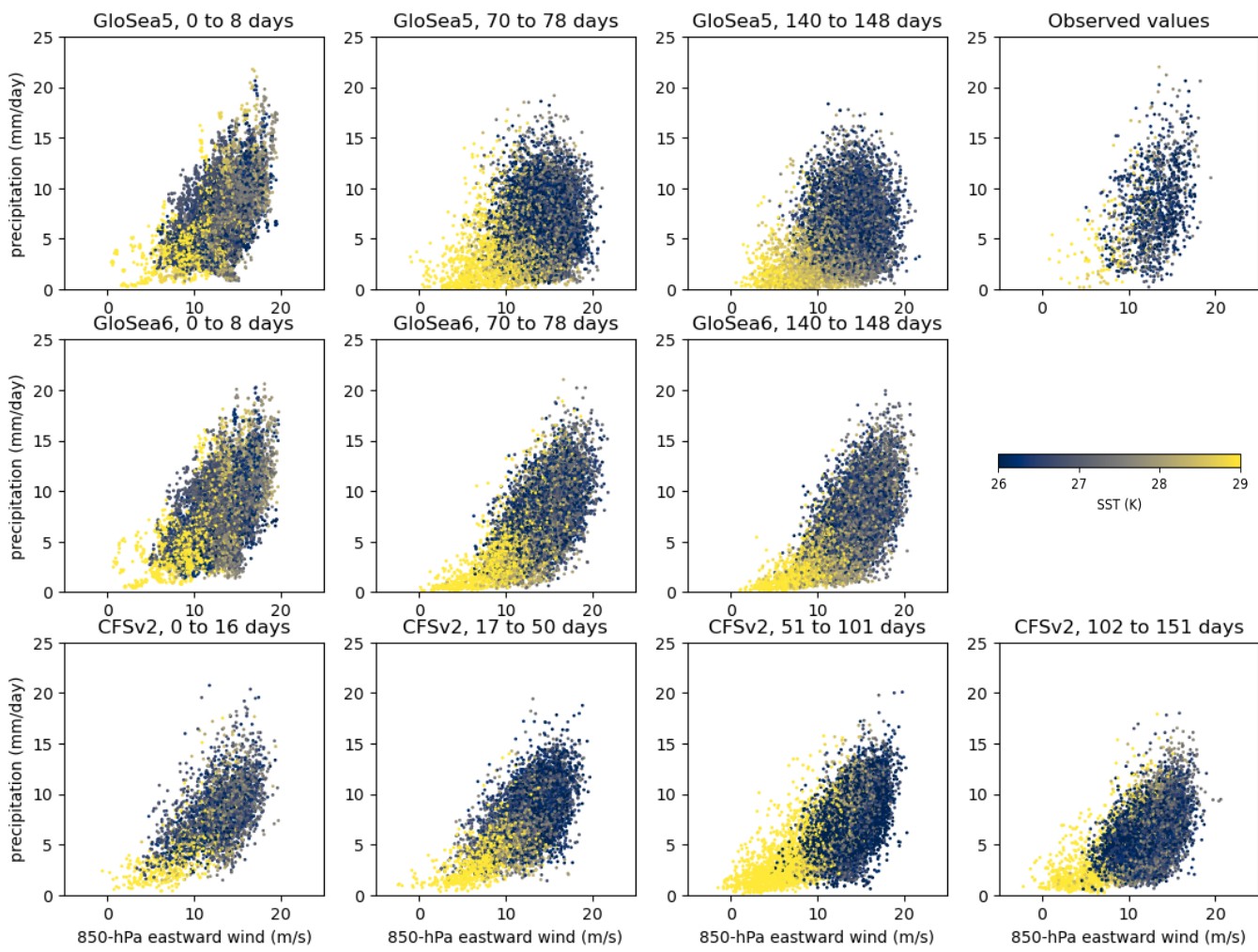

**Figure 8. Scatter plots of precipitation over India (8–29N, 69–89E) against 850-hPa eastward wind over the Arabian Sea (0–21N, 49–69E), coloured by SST over the Arabian Sea. Each dot represents a single hindcast for one year during 2002–2015, one day in June, July or August and one lead time within the range shown.**

### 3.2 Behaviour during different parts of the monsoon season

The development with lead time for different parts of the monsoon season is shown in Fig. 9. Because the CFSv2 hindcasts are only available initialised from February, the longest lead times are not available in June and July. All three months show sharp initial decreases. In June this is followed by a consistent but decelerating decrease throughout the length of the hindcast. The recovery followed by a further decrease seen for CFSv2 is present in all months, but is much stronger in July and August. The less substantial recovery behaviour seen for GloSea also occurs in all months, but in August on somewhat longer time scales: the worst low-precipitation biases are seen for lead times of 8–50 days, with the bias reducing to a smaller value from

50 to 100 days. It is also notable that the differences between the three model setups are much larger in July and August than in June.

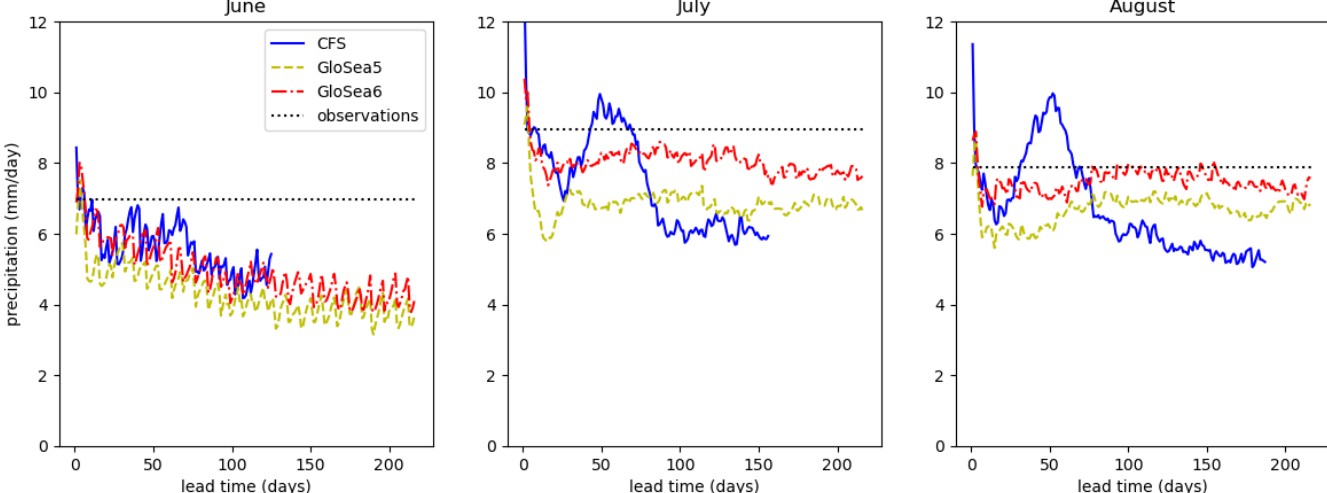

**Figure 9. Variation of precipitation with hindcast lead time (averaged over 8N–29N, 69E–89E and 2002–2015) for hindcasts valid in each of June, July and August. The dotted lines show observed values averaged over all dates in the month (i.e., not just dates on which hindcasts are available).**

The behaviour in different parts of the season in GloSea is consistent with the findings of Martin & Levine (2012), who evaluated the seasonal cycle of precipitation over a similar region to that used in this study for climate simulations using an earlier version of the GC model. Looking at the recent past climate, they found both atmosphere-only (forced by observed SSTs) and coupled-model simulations to produce too little precipitation over the region throughout June–August, but with the bias being worse in the coupled simulation earlier in the period, and worse in the atmosphere-only simulation later in the period. The early poorer performance of the coupled simulation was attributed to a delayed monsoon onset caused by cold SST biases over the Arabian Sea, as described by Levine & Turner (2012). Meanwhile, the later better performance of the coupled simulation was attributed to a cold SST bias over EIO associated with reduced precipitation there, leading to increased precipitation over the Indian Peninsula.

Equivalent plots to Fig. 7, but restricting to each of June, July and August, are shown in Figs. 10–12. The initial decrease in precipitation is accompanied by an increase in precipitation over the EIO in all months. In GloSea, this is particularly strongly tied to the flow from India to the EIO in July and August. The subsequent reduction in precipitation over the EIO is slower in August and is accompanied by a later recovery in the precipitation over India. The initial increase in eastward wind over the Arabian Sea is weaker in June in all models, and the flow itself is generally weaker as the monsoon has not yet fully developed. The flow subsequently decreases in June, accompanied in GloSea by decreasing SSTs.

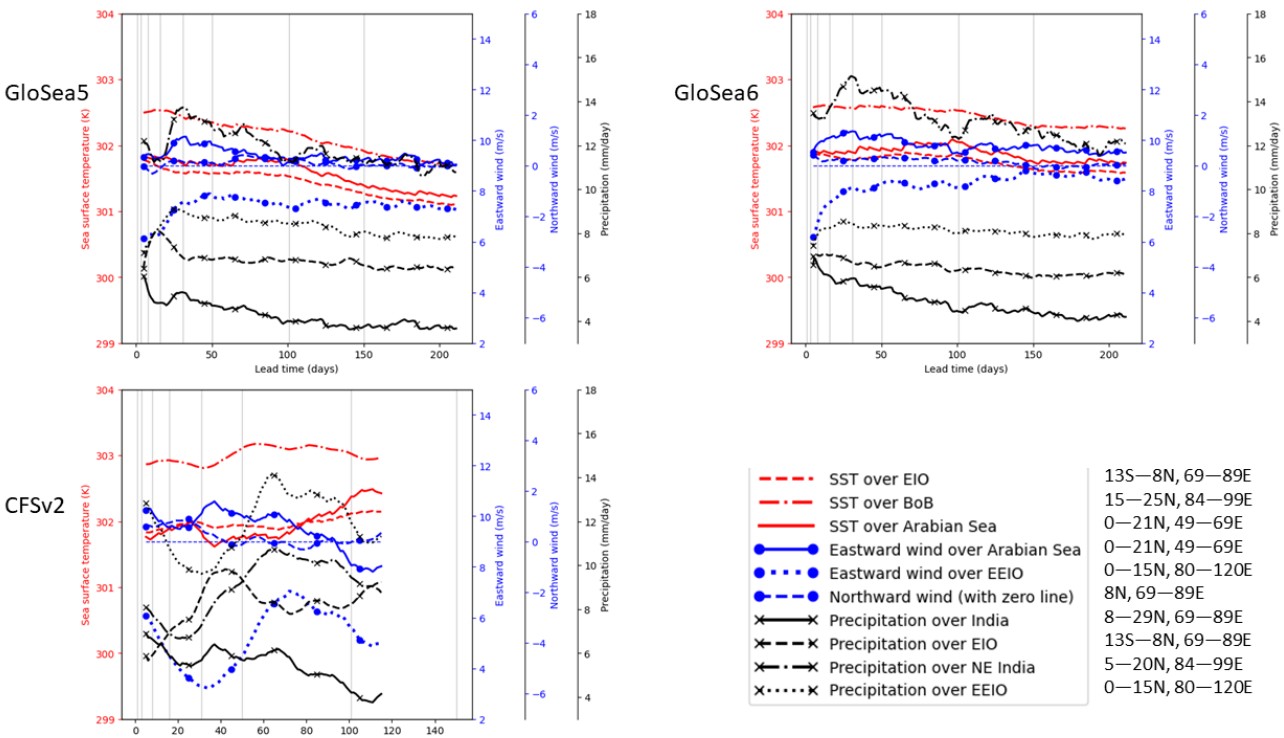

**Figure 10: SST, 850-hPa wind and precipitation averaged over different regions (shown in Fig. 5), averaged over June 2002–2015, as a function of lead time. A smoothing is applied by taking a 9-day running average in the lead-time direction.**

In CFSv2, it is evident that the recovery in precipitation over EIO is much stronger for July and August. In contrast, the recovery is smaller for June, and it occurs much later (~70 days of lead time). This might be because the bias in moisture transport over EIO is not very large in June, as reported by Napusetty et al. (2016). For July and August, there is a rapid increase in the zonal flow over the EIO, such that the precipitation also increases rapidly, peaking at ~50–60 days lead time. This branch of zonal winds, which feeds moisture to EIO, likely pulls away the moisture from the Arabian Sea branch, such that the precipitation over India declines as the zonal flow builds up rapidly and peaks over the EIO region. This argument is supported by the fact that the rate of build-up of zonal flow over EIO during July and August is much more rapid than that of the Findlater jet.

The biases in moisture transport over EIO appear to contribute significantly to India's biased annual cycle of precipitation. The peak in monsoon rainfall occurs during July in observations. However, CFSv2 has a relatively flat annual cycle of precipitation over India, wherein the maximum rainfall occurs during August, but the difference between July and August rainfall is small (Ramu et al. 2016). The bias in moisture transport over EIO, its lead-time dependence, and its contribution in pulling moisture away from the Indian landmass is therefore important, and will be investigated in future studies.

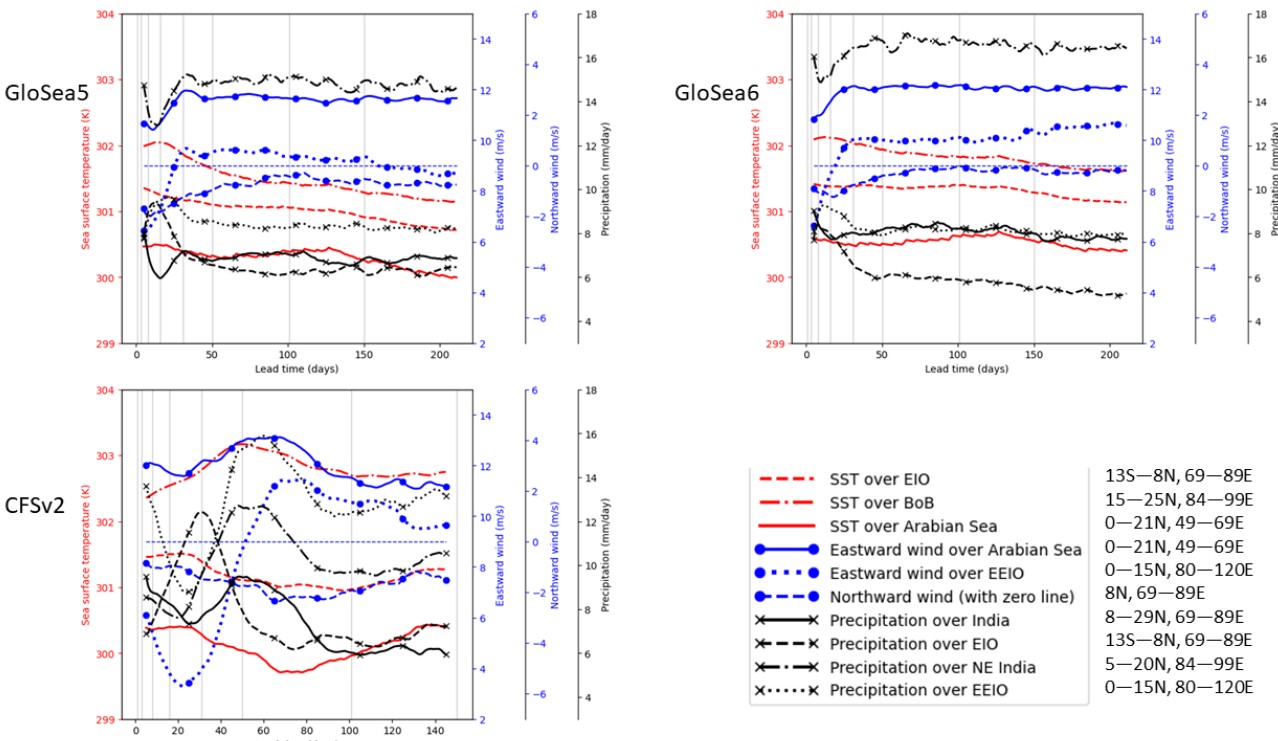

**Figure 11. SST, 850-hPa wind and precipitation averaged over different regions (shown in Fig. 5), averaged over July 2002–2015, as a function of lead time. A smoothing is applied by taking a 9-day running average in the lead-time direction.**

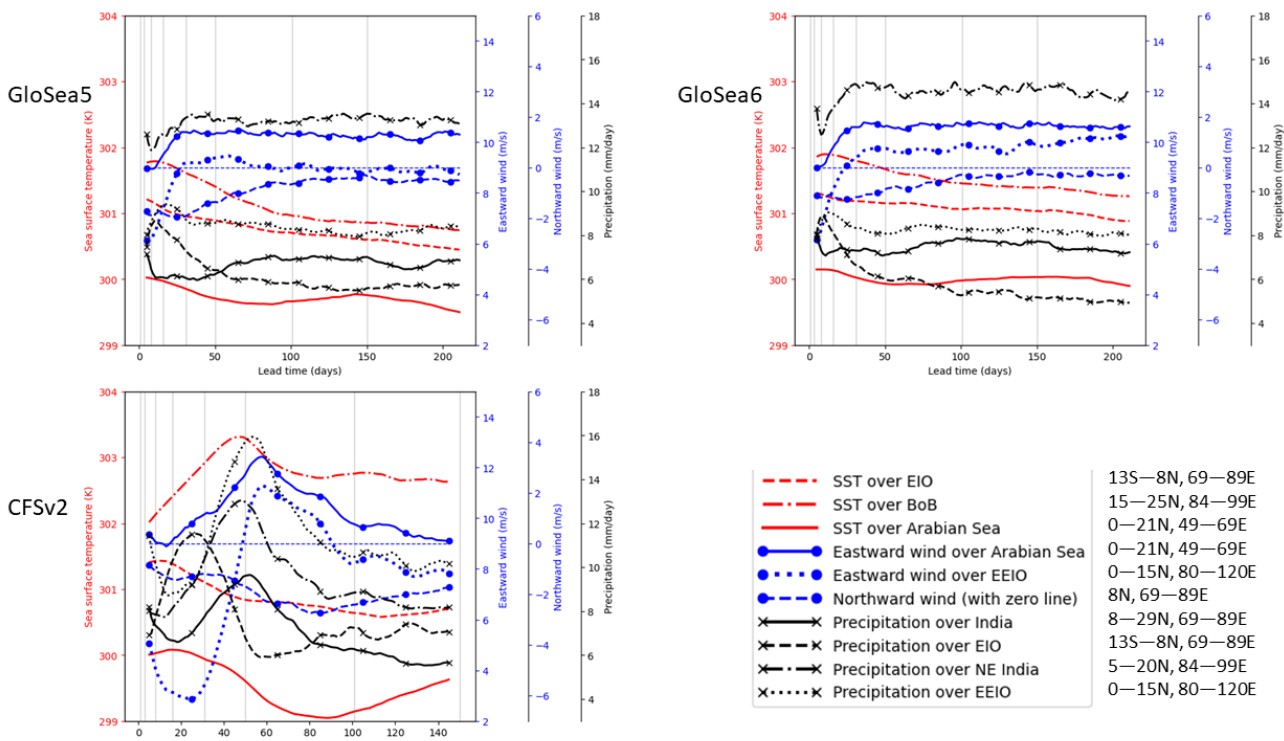

**Figure 12. SST, 850-hPa wind and precipitation averaged over different regions (shown in Fig. 5), averaged over August 2002–2015, as a function of lead time. A smoothing is applied by taking a 9-day running average in the lead-time direction.**

## 3.3 Response to large-scale drivers

### 3.3.1 Variation with BSISO phase at hindcast valid time

The hindcasts are categorised according to the observed BSISO phase at the valid time, following Keane et al. (2021), taking precipitation averaged over latitudes 8N–29N and 69E–89E. For each lead time, all hindcasts where the observed BSISO

amplitude at valid time is greater than 1 are assigned a phase equal to the observed phase on the date at that lead time (those with amplitude less than or equal to 1 are discarded for this method). For each phase, an average is taken of all the area-averaged precipitation values, to produce a quantity that varies as a function of BSISO phase and lead time. This quantity is plotted as a function of phase in Fig. 13, for selected lead times, with a further 8-day average over lead time to reduce noise. Also plotted is the quantity for observations, which applies the same method (including the restriction to BSISO amplitudes

greater than 1) to observed precipitation over all days during June–August 2002–2015. Versions of Fig. 13 using fewer GloSea members and using a longer range of GloSea years are shown in Appendix A (Figs. A12 and A13, respectively), and are very similar.

Looking at GloSea5 during the first 8 days, the bias is worse for phases where the large-scale dynamics implies increasing precipitation and less bad when the large-scale dynamics implies decreasing precipitation, in agreement with the findings of

Keane et al. (2021) for 7-day forecasts. The situation is slightly different for GloSea6 and CFSv2, with the precipitation generally still too high when averaged over the first 8 days of the hindcast, but there is the same shift in the peak precipitation from phase 4 in the observations to phase 5 in all the models.

Comparing the second 8 days with the first 8 days for all models, it is clear that the behaviour continues, with generally a reduction for all phases, but a stronger reduction when the large-scale dynamics implies increasing precipitation and vice-

versa. From day 16 this continues but becomes weaker as the hindcasts lose their phase dependence until at day 40 the errors

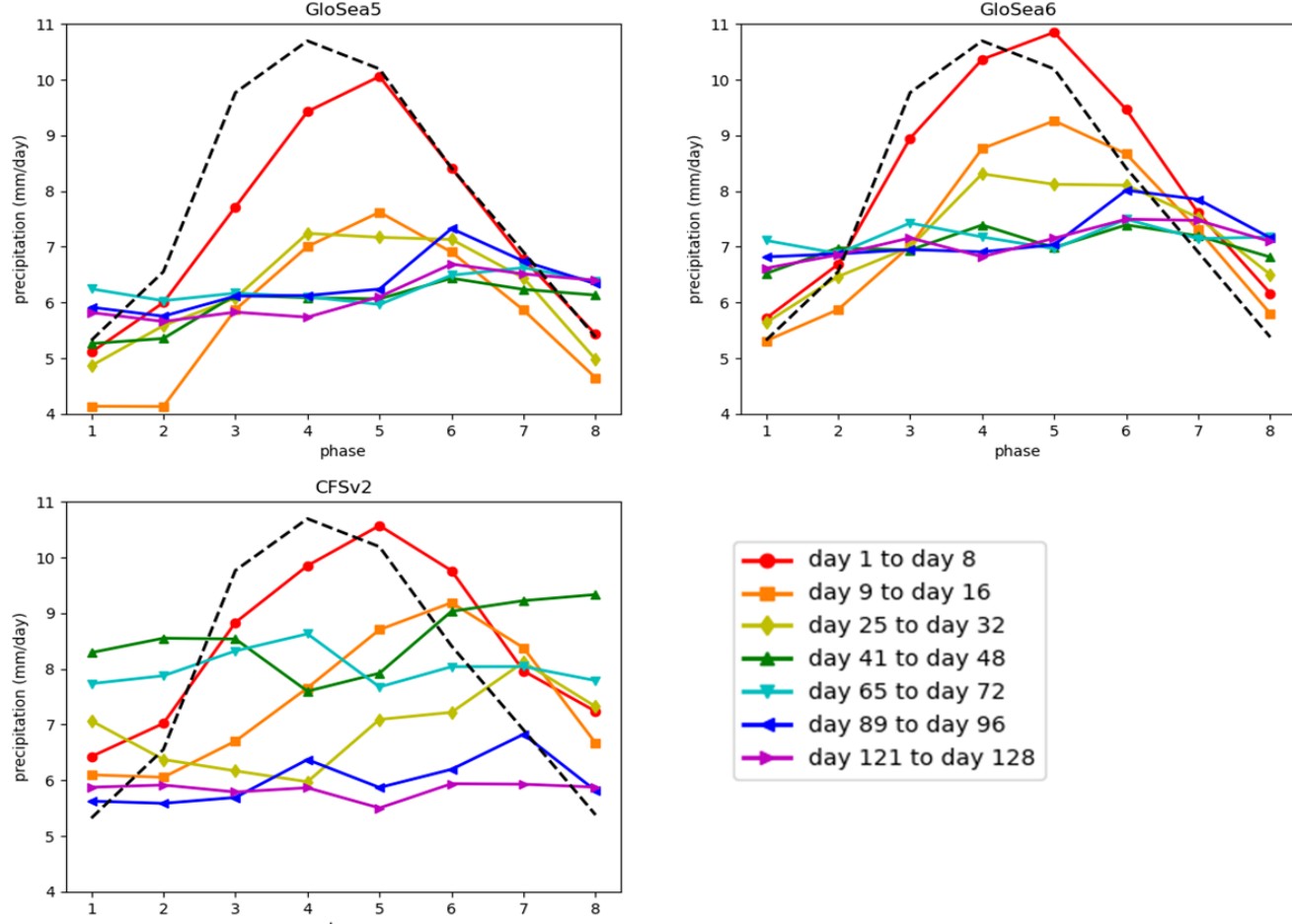

**Figure 13: Variation in precipitation (averaged over 8N–29N, 69E–89E and June–August 2002–2015) with lead time as a function of observed BSISO phase at valid time, averaged over 8-day lead time blocks (coloured lines). Each panel corresponds to a model denoted by its title. Observed precipitation corresponding to each BSISO phase is denoted by the black dashed line in all three panels.**

are essentially independent of phase. This suggests that the models lose their capability to forecast the BSISO phase by 40 days into the hindcast, as the model precipitation (and, therefore, any BSISO phase that it is simulating) has no significant

relationship with the observed BSISO phase. This is consistent with previous work that has shown that seasonal forecast models can effectively simulate the BSISO on time scales of tens of days (e.g., Lee et al., 2015; Fang et al., 2019).

### 3.3.2 Variation with BSISO phase at initial time

In order to establish the role of initial conditions on the development of the rainfall biases over India in the two seasonal forecast systems, hindcasts initialised from 1st June to 1st August (inclusive), for each year, are next categorised according to

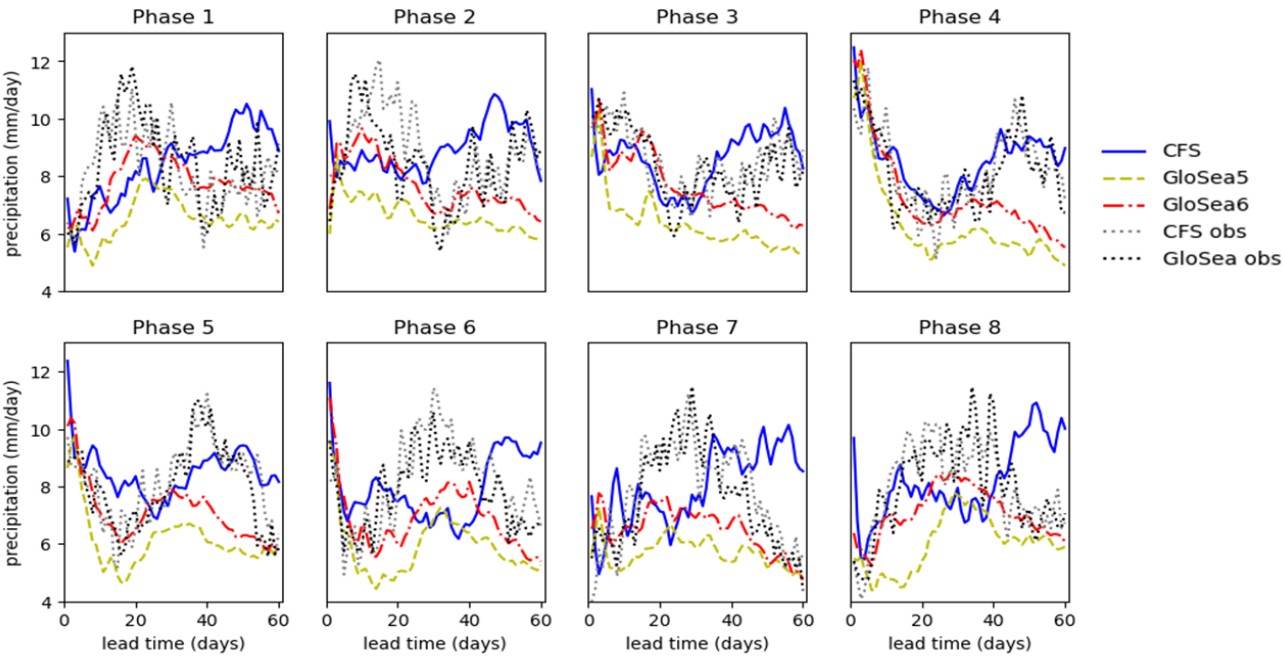

**Figure 14: Variation in precipitation with hindcast time, for hindcasts initialised during each of 8 BSISO phases. Values are averaged over 8N–29N and 69E–89E and over hindcasts initialised during the period 1st June to 1st August 2002–2015 on dates where the observed BSISO was in the given phase. The black dotted lines show observed values averaged over the same dates as the hindcasts at the relevant lead time.**

the observed BSISO phase at the start of the hindcast. The precipitation is again averaged over latitudes 8N–29N and 69E–89E, and over all hindcasts corresponding to each phase, and values for the first 60 hindcast days in each system are plotted in Fig. 14. For this method there is no temporal coarse-graining into 8-day blocks; therefore, no minimum restriction on the BSISO amplitude is applied, so that a larger data sample can be taken to reduce random temporal fluctuations that would otherwise be reduced by the 8-day coarse-graining. The dotted lines show observed values averaged over the same dates on which each length of hindcast is available; as for Fig. 2, this is different between the two systems as the hindcasts are initialised on different dates. Figure A14 shows a similar set of plots, but also including hindcasts initialised in the rest of August, and extending only to 30 days: these plots from a somewhat larger sample size are in broad agreement with the first 30 days of those in Fig. 14.

We use the observed curves to define how the large-scale drivers affect the precipitation through the course of each hindcast, and they are similar to what is expected from the standard dynamical analysis that is used to define the phases. For example, hindcasts starting in phase 4 have high initial precipitation on average, and this decreases early in the hindcast, while those starting in phase 8 have low initial precipitation on average, and this increases early in the hindcast. Moreover, the curves follow an oscillation in precipitation with a period of about 40 days, corresponding to the period of the BSISO cycle.

A general conclusion from Fig. 14 is that both models perform better when the large-scale dynamics is driving decreases in precipitation than when it is driving increases in precipitation. For example, the first 30 days are well simulated when the hindcast starts in phases 4 or 5, whereas the precipitation is far too low over the first 30 days when the hindcast starts in phases 8, 1 or 2, particularly for GloSea5 (CFSv2 sometimes even simulates too much precipitation, although this is usually associated with an initial high-precipitation bias). This continues further into the hindcast in GloSea: for phases 4 and 5, although the first 30 days are well simulated, the subsequent observed increasing precipitation is not well captured, and the improvement in GloSea6 compared to GloSea5 in capturing increasing precipitation is less than that during the first 30 days of the hindcast (e.g., phases 8, 1, 2). CFSv2 does simulate the later increase for phases 4 and 5, although this could be because it produces increasing precipitation more generally during days 30 to 50 of the hindcast.

The variation in model precipitation development with BSISO phase may be quite different in the bias-corrected GloSea seasonal forecasts from the behaviour seen here in the hindcasts. Intuitively, given that the bias-correction depends only on start date and lead time (Arribas et al., 2011; MacLachlan et al., 2015), it may be expected that the forecasts would have a positive precipitation bias for phases 4 and 5 and a negative bias for phases 1 and 8. It would be interesting to investigate whether this is indeed the case, and whether it could be improved by using some information about expected BSISO phases, at least for shorter lead times.

## 4 Discussion

This study investigates ISM biases in two seasonal forecasting systems, CFSv2 and versions 5 and 6 of GloSea. Both systems initially have a reduction in precipitation with increasing lead time, accompanied by increasingly anti-cyclonic flow, and in GloSea the precipitation reduction is shown to fully occur within the first 8 days of the hindcast. This corroborates previous work (e.g., Rodwell and Palmer, 2007; Martin et al., 2010; Rodriguez and Milton, 2019; Martin et al., 2021) showing that future work studying biases in weather forecasts (and, more generally, short-time-integration simulations) will provide substantial insight into biases across timescales, without the requirement to conduct lengthy climate simulations, particularly for the GC system. A strong focus for this work should be investigating the cause of the increasing precipitation over the EIO at short lead times: a link between excessive convergence over the EIO and low-precipitation biases over India has previously been identified both in GC (Bush et al., 2015) and in CFSv2 (Shukla and Huang, 2016), and Martin & Rodriguez (2024) demonstrated a change in behaviour between the first 10 days and longer lead times over the eastern EIO that is less apparent in the western EIO. Both systems show a substantial decrease within a lead time of 25 days, and the results for GloSea are in

broad agreement with previous work on atmosphere-only weather forecasts. This suggests that issues in the atmospheric model component of the system are sufficiently important that atmosphere-only simulations can continue to play a role in investigating these biases.

It is notable that GloSea6 has improved (i.e., smaller) biases compared to GloSea5. Although the ISM low-precipitation bias is an important factor in evaluating the performance of GC3 compared with GC2, improvements in GC versions are aimed at Global model performance as a whole and on the full range of time scales, from weather forecasting to climate simulation. It is therefore encouraging that the combined effect of these improvements on the simulation of the ISM on seasonal time scales has been so substantially positive. The upgrade from GC2 to GC3 (Williams et al., 2017) includes a wide range of improvements to the atmosphere and land components (Walters et al., 2017), the ocean component (Storkey et al., 2017) and the sea ice component (Ridley et al., 2017). In particular, the previous upgrade in the atmospheric component (GA6) had focused on the dynamical core, with changes to the physics parameterisations relatively restricted, so that the upgrade from GA6 (used in GC2) to GA7 (used in GC3) included a relatively large number of substantial longer-term changes to the atmospheric parameterisation schemes. Given that the improvement is present within 8 days, it is likely that upgrades to the atmospheric component are largely responsible for the improved performance in GloSea6. Walters et al. (2017) show that GA7 does have reduced summer precipitation biases over India compared with GA6, and attribute this to improvements in the stochastic physics forcing, an upgrade in the convection parameterisation from 5A to 6A and improved warm rain microphysics. Improvements in the scale adaptivity of the model play a large role in these upgrades: Sanchez et al. (2016) show that the inclusion of a resolution-dependent factor for the convection dissipation rate produces higher kinetic energy perturbations at lower resolutions, leading to reduced tropical biases, and the 6A convection scheme is designed to be effective at higher vertical resolutions than those for which the 5A convection scheme was designed.

After its initial reduction, the precipitation recovers in CFSv2, so that it is in agreement with observations over lead times of 30–70 days. It could therefore be interesting to study this recovery period further: although it is based on errors in the model (as quantities should not vary with forecast/hindcast lead time as plotted in Fig. 2), this could provide insight into conditions under which the model is capable of simulating break-to-active transitions. A similar, albeit much smaller and more short-lived, recovery is also identified in GloSea, suggesting that such insight could apply to different modelling systems. In this study, the recovery has been shown to be associated with increasing westerly flow over the Arabian Sea in both modelling systems.

Although the biases seem to develop from issues in the atmospheric model components, the interaction with SSTs in the ocean component does play some role, and it is likely that this role becomes more important beyond the seasonal time scale, as biases in climate simulations are generally larger than those demonstrated here in the seasonal hindcasts. In CFSv2 the interaction seems to be quite direct (with decreasing SSTs corresponding closely with decreasing precipitation), and more important than in GloSea, where it seems to depend more on the part of the season that is evaluated. This could be due to differences in the two ocean model components or in the coupling between the atmosphere and the ocean; Bollasina and Nigam (2009) showed that different coupled models exhibit varyingly deficient representation of local and non-local air–sea interactions in the Indian

Ocean during boreal summer and, in particular, that they tend to overestimate the correlation between SST and precipitation, suggesting that local air–sea interactions are overemphasised. Meanwhile, when focussing on June, the behaviour of the two systems (and the two versions of GloSea) is much more similar, and suggests that both systems suffer from a delayed monsoon onset, so this could be a common issue affecting both systems in the same way.

An analysis of how the forecast precipitation depends on observed BSISO state has been carried out in this study, in terms of both the phase at the beginning of the forecast and the phase at the end of the forecast. Both evaluation methods show that the two systems are best at simulating situations where the large-scale dynamics favours decreasing precipitation over India, and are worst at simulating situations where the large-scale dynamics favours increasing precipitation over India, and that this continues beyond the first 8 days of the forecast and even into a second BSISO cycle. This suggests an opportunity to focus future work on cases corresponding to such increasing precipitation conditions, and it will be interesting to investigate how widespread the behaviour is amongst other models. It also provides useful information to users on the relative reliability of forecasts of each of these transitions.

One explanation for the dependence on BSISO phase could be that, based on their systematic biases, the two systems have a tendency to move towards monsoon break conditions and so are better at capturing situations where this is occurring in reality. Further analysis of such transitions in the models may provide insight into the reasons for this preference and its contribution to the overall systematic biases in ISM rainfall in climate models. Gera et al. (2021) evaluated seasonal forecasts of the ISM using the NCMRWF-ERP system based on GC2 and found that, while break-to-active and active-to-break transitions were both predicted well up to 4 weeks, there was some evidence of a weakening and a delay in the break-to-active transitions with increased forecast lead time.

The BSISO analysis also shows that the precipitation in both systems is largely independent of observed BSISO state by about 40 days into the forecast. It will therefore be interesting in future work to carry out a similar analysis of how the biases vary with the models' own BSISO state and, indeed, to what extent the models are able to simulate the BSISO. The models may simulate different distributions of precipitation as a function of their own BSISO phase and this could change as the lead time increases. It may also be useful to use the longer-range predictability of the BSISO phase to add information about likely biases in weather and seasonal forecasts, particularly if the BSISO is not well simulated by the models, in which case this information would not already be included in the forecasts themselves. Previous studies have shown that the BSISO can be predicted in forecast models up to four weeks in advance (Jie et al., 2017; Xiang et al., 2024) and this could be combined with statistical processing to obtain a longer-range prediction of likely model biases.

The interannual variation of the ISM is affected by both the El Niño-Southern Oscillation (ENSO; Krishnamurthy & Goswami, 2000; Chattopadhyay et al., 2015; Xavier et al., 2007) and the Indian Ocean Dipole (IOD; Pothapakula et al., 2020; Hrudya et al., 2021). Ashok et al. (2001) showed that ENSO and the IOD can have complementary effects on the monsoon, with high ENSO-rainfall correlations accompanied by low IOD-rainfall correlations and vice-versa. It will therefore be interesting to investigate whether the relationship between rainfall bias and BSISO phase varies depending on the indices of ENSO and the

IOD. Kikuchi (2020; 2021) showed that ENSO has little effect on the BSISO overall, but it can affect certain aspects of the BSISO (Wu & Cao, 2017; Li & Mao, 2019) and the Monsoon Intraseasonal Oscillation (Joseph et al., 2011).

Lee et al. (2013) attribute the BSISO2 indices, corresponding to the 3rd and 4th EOFs, to being relevant to the pre-monsoon and onset phase, so an evaluation of the relationship between precipitation biases and BSISO2 phases (the present study evaluates

this relationship for BSISO1 phases, corresponding to the 1st and 2nd EOFs, and referred to here simply as "BSISO phases") would be worthwhile. In the present study, it is found that the biases have different characteristics in June from those in July and August, so it may be that applying the BSISO analysis to the months separately, and with the two indices separately, will identify further relationships between biases and BSISO phases.

As already mentioned, the purpose of seasonal forecasts is largely to produce a statistical idea of the state of the weather a few

weeks to months ahead, and the hindcasts evaluated in this study are in practice used to calibrate the actual forecast models, so that systematic biases should not directly affect the quality of the forecast. However, it may be of interest in future work to investigate whether there is any relationship between model bias and forecast skill in the seasonal forecasts.

**Appendix A: Additional figures and table**

| Initialization month | Dates |
|---|---|
| February | 05, 10, 15, 20, 25 (00 &12 UTC) |
| March | 02, 07, 12, 17, 22, 27 (00 & 12 UTC) |
| April | 01, 06, 11, 16, 21, 26 (00 & 12 UTC) |
| May | 06, 11, 16, 21, 26 (00 & 12 UTC) |
| June | 05, 10, 15, 20, 25 (00 &12 UTC) |
| July | 05, 10, 15, 20, 25 (00 &12 UTC) |
| August | 05, 10, 15, 20, 25 (00 &12 UTC) |

**Table A1. Start dates of CFSv2 hindcasts evaluated in this study.**

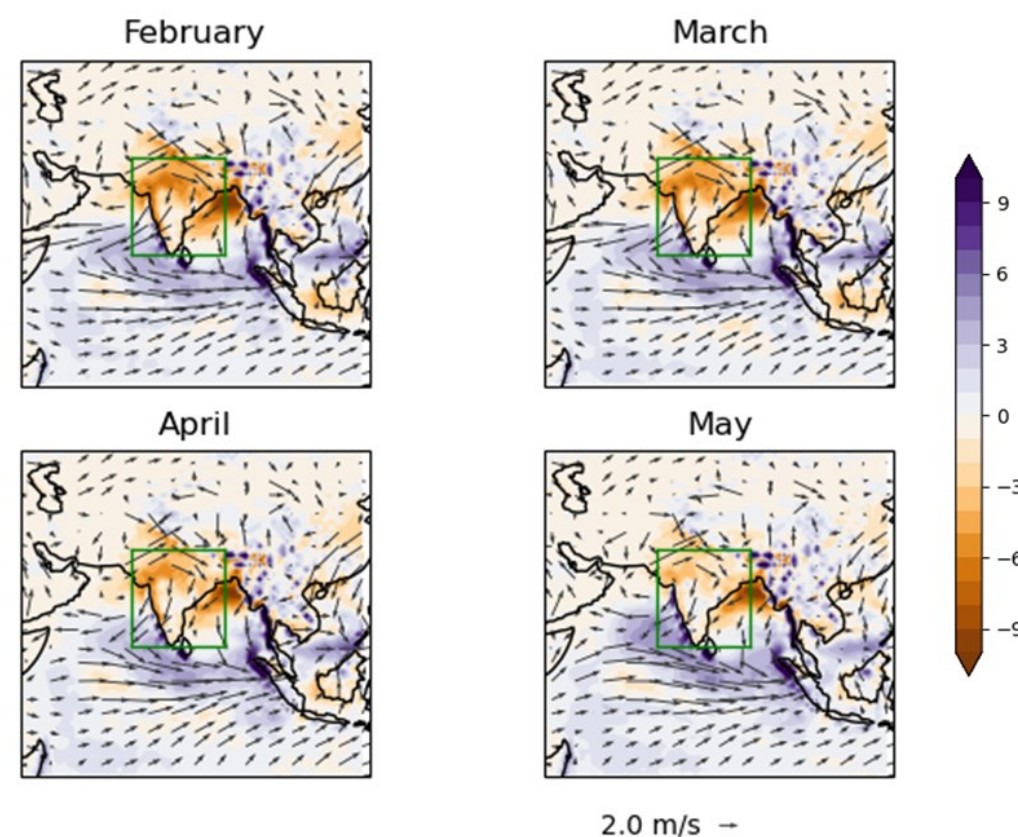

**Figure A15. Overall CFSv2 biases averaged over valid times in June–August 2002–2015 and initial times in the month shown. Variables are precipitation in mm/day (bias against IMERG observations) and 850-hPa horizontal wind (bias against ERA5 reanalysis).**


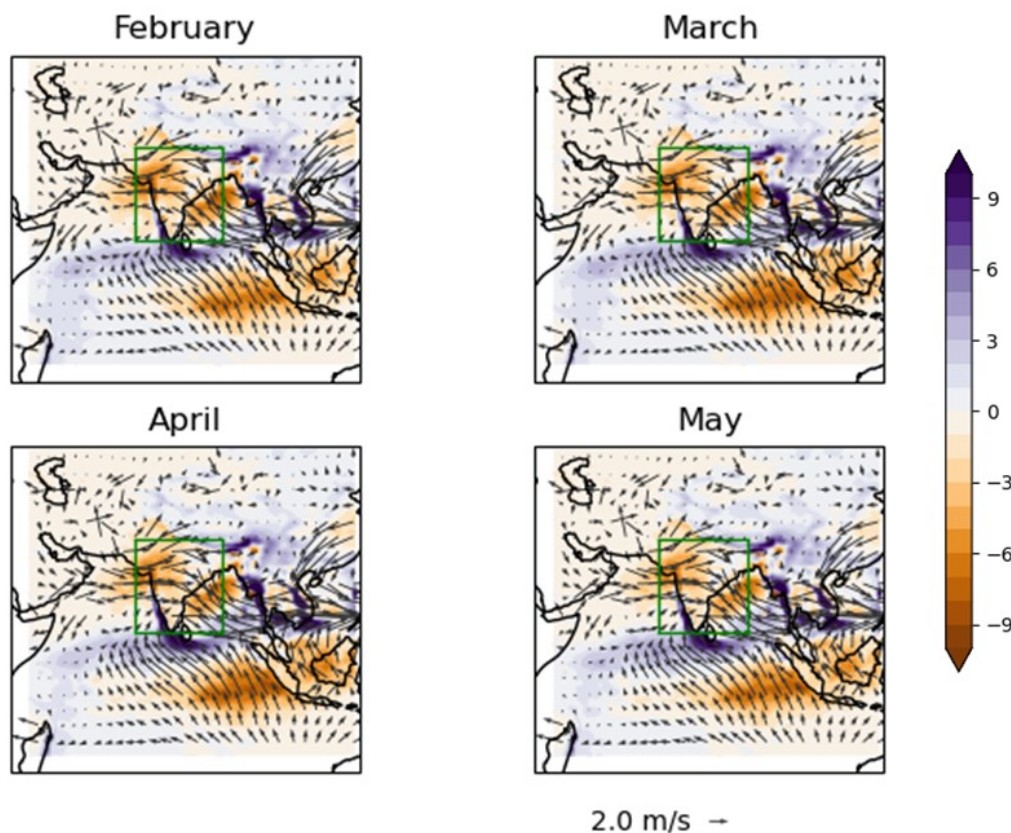

**Figure A16. Overall GloSea5 biases averaged over valid times in June–August 2002–2015 and initial times in the month shown. Variables are precipitation in mm/day (bias against IMERG observations) and 850-hPa horizontal wind (bias against ERA5 reanalysis).**


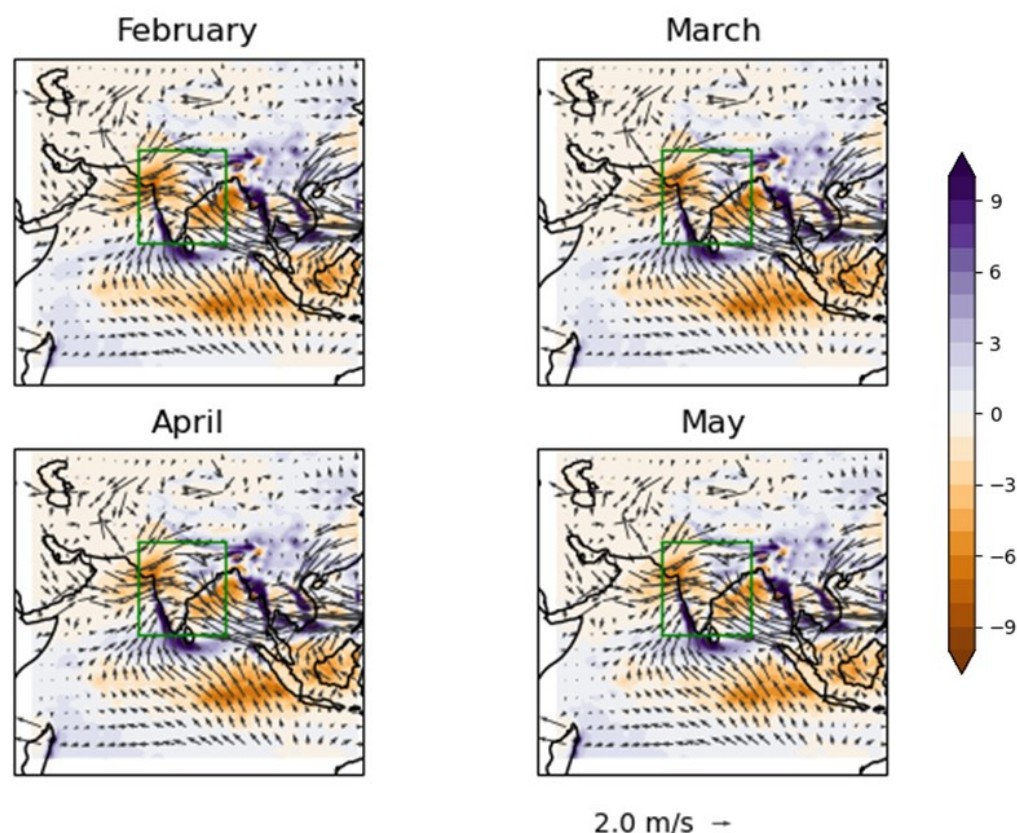

**Figure A17. Overall GloSea6 biases averaged over valid times in June–August 2002–2015 and initial times in the month shown. Variables are precipitation in mm/day (bias against IMERG observations) and 850-hPa horizontal wind (bias against ERA5 reanalysis).**


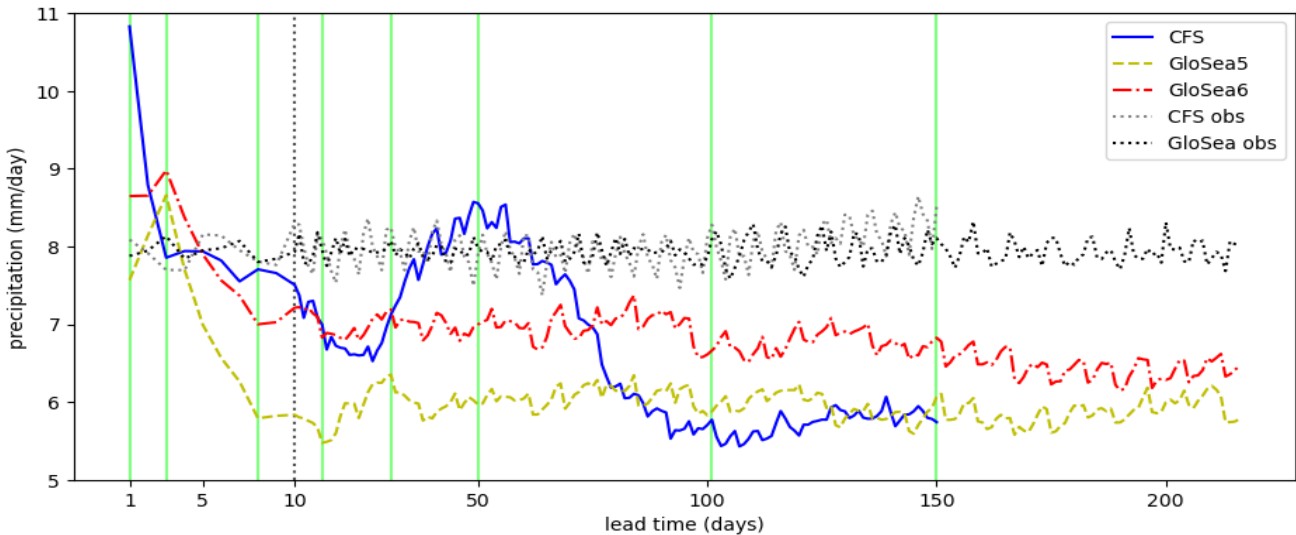

**Figure A4. As Fig. 2 but with only 3 of the 7 ensemble members in GloSea.**

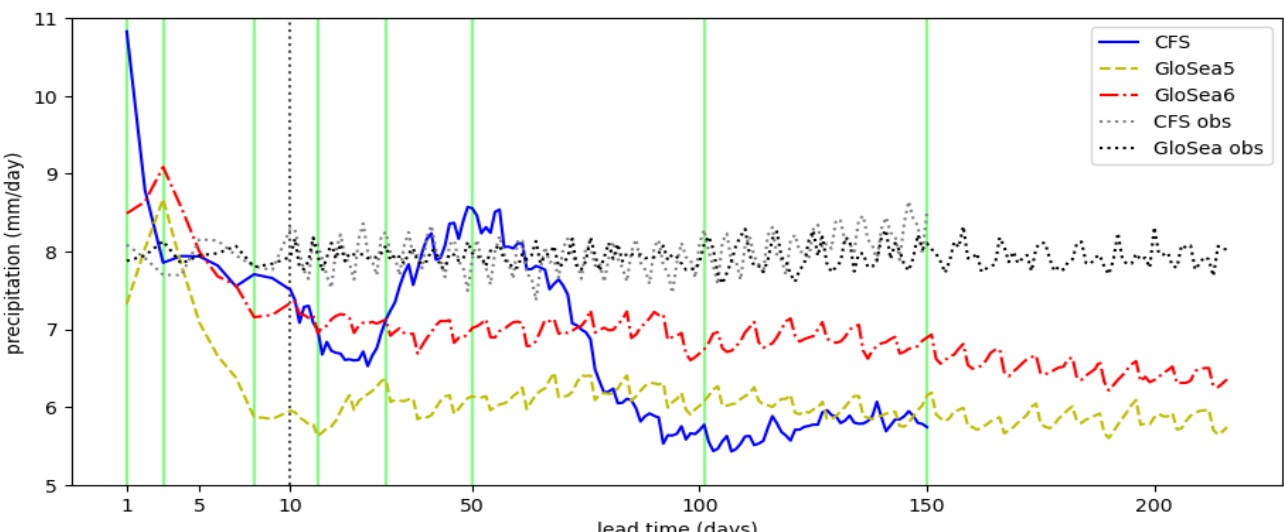

**Figure A5. As Fig. 2 but with GloSea5 hindcasts valid during 1994–2015 and GloSea6 hindcasts valid during 1994–2016 (all other data unchanged, i.e., "GloSea obs" still refers to the period 2002–2015).**


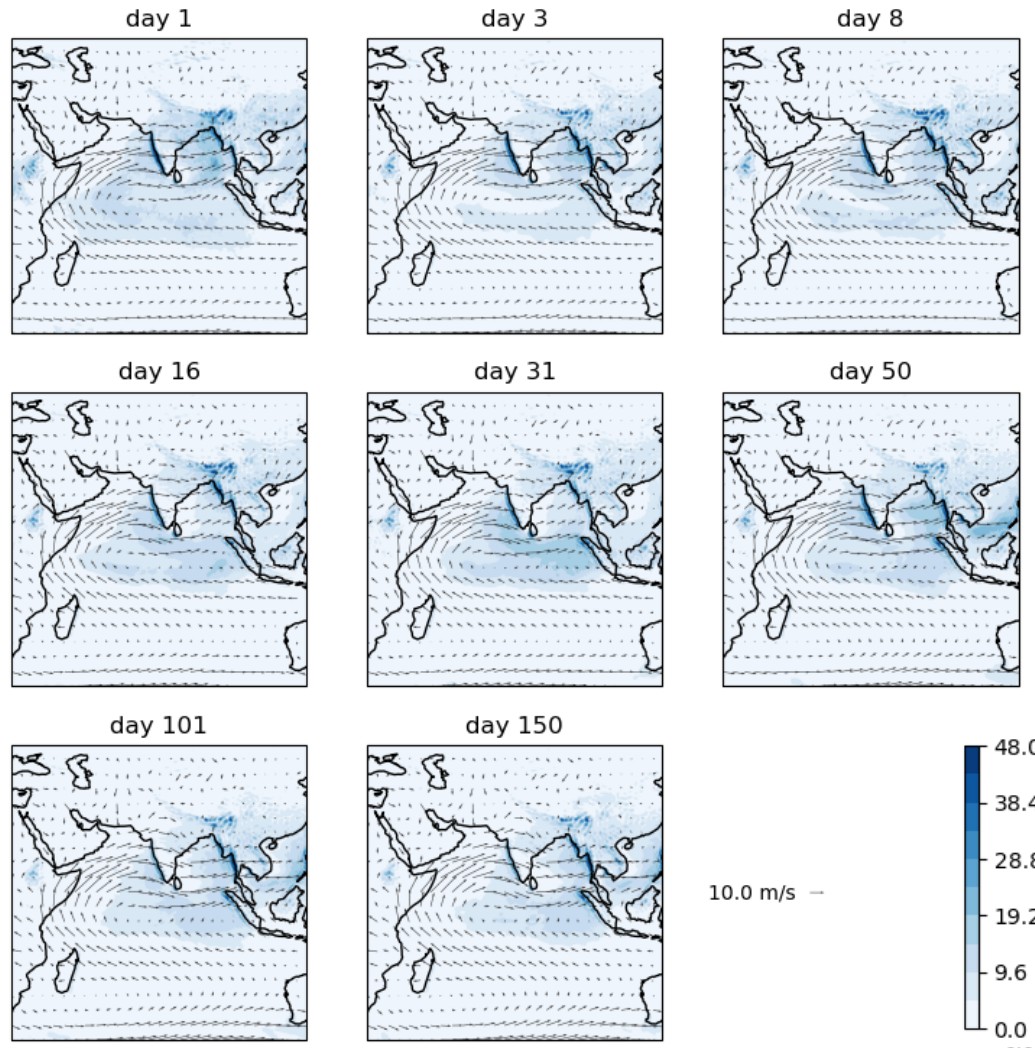

**Figure A6. Precipitation (colours, in mm/day) and 850-hPa wind (vectors) in CFSv2, at the hindcast lead time shown, averaged over June–August 2002–2015.**

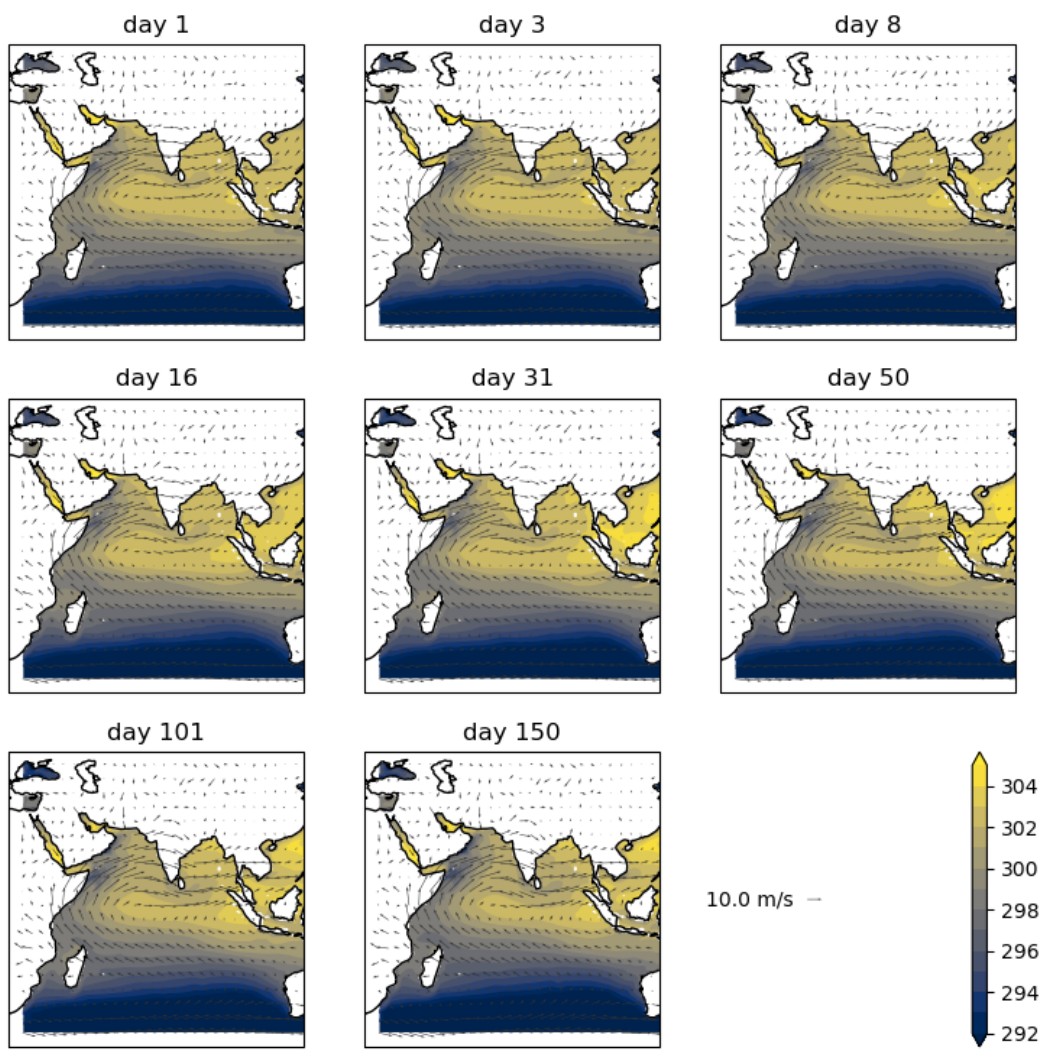


**Figure A7. SST (colours, in K) and 850-hPa wind (vectors) in CFSv2, at the hindcast lead time shown, averaged over June–August 2002–2015.**

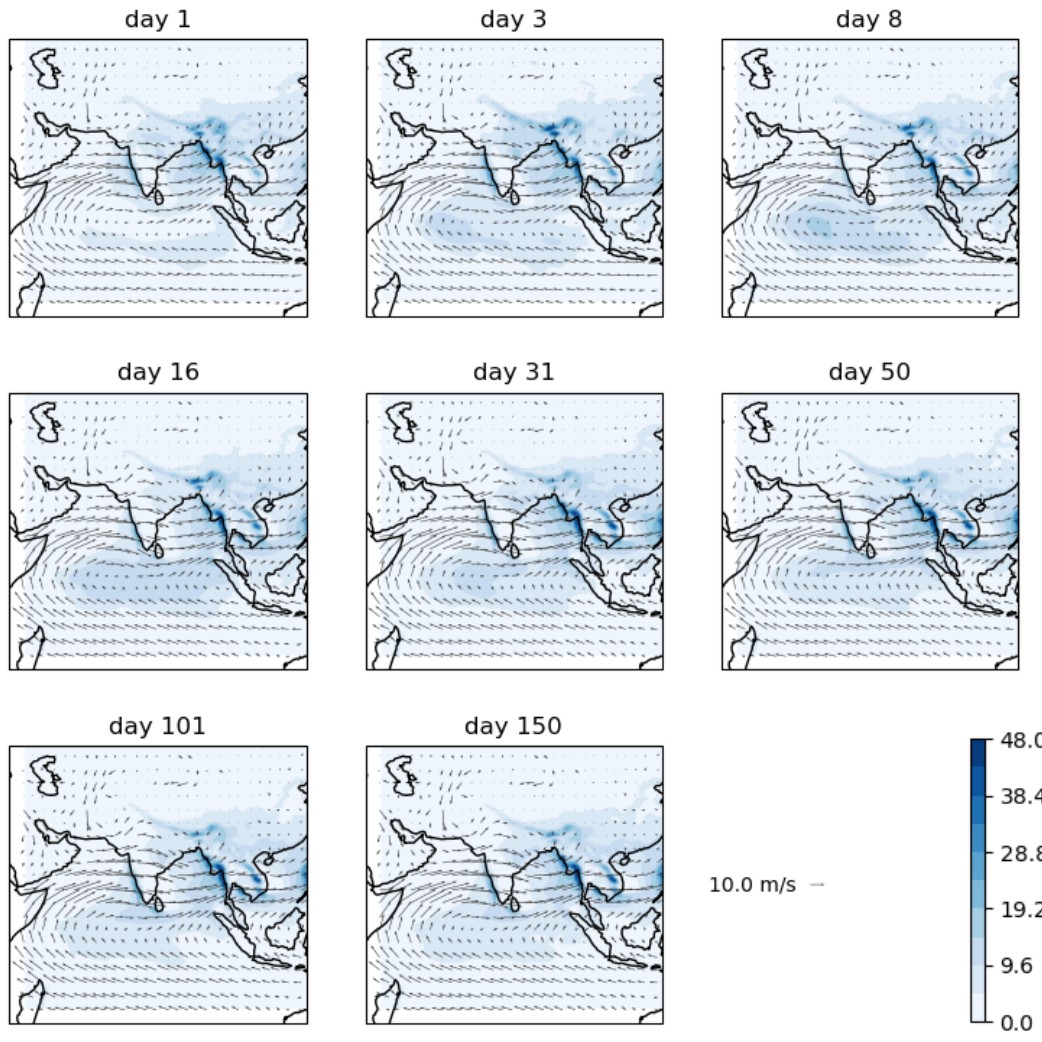

**Figure A8. Precipitation (colours, in mm/day) and 850-hPa wind (vectors) in GloSea5, at the hindcast lead time shown, averaged over June–August 2002–2015.**

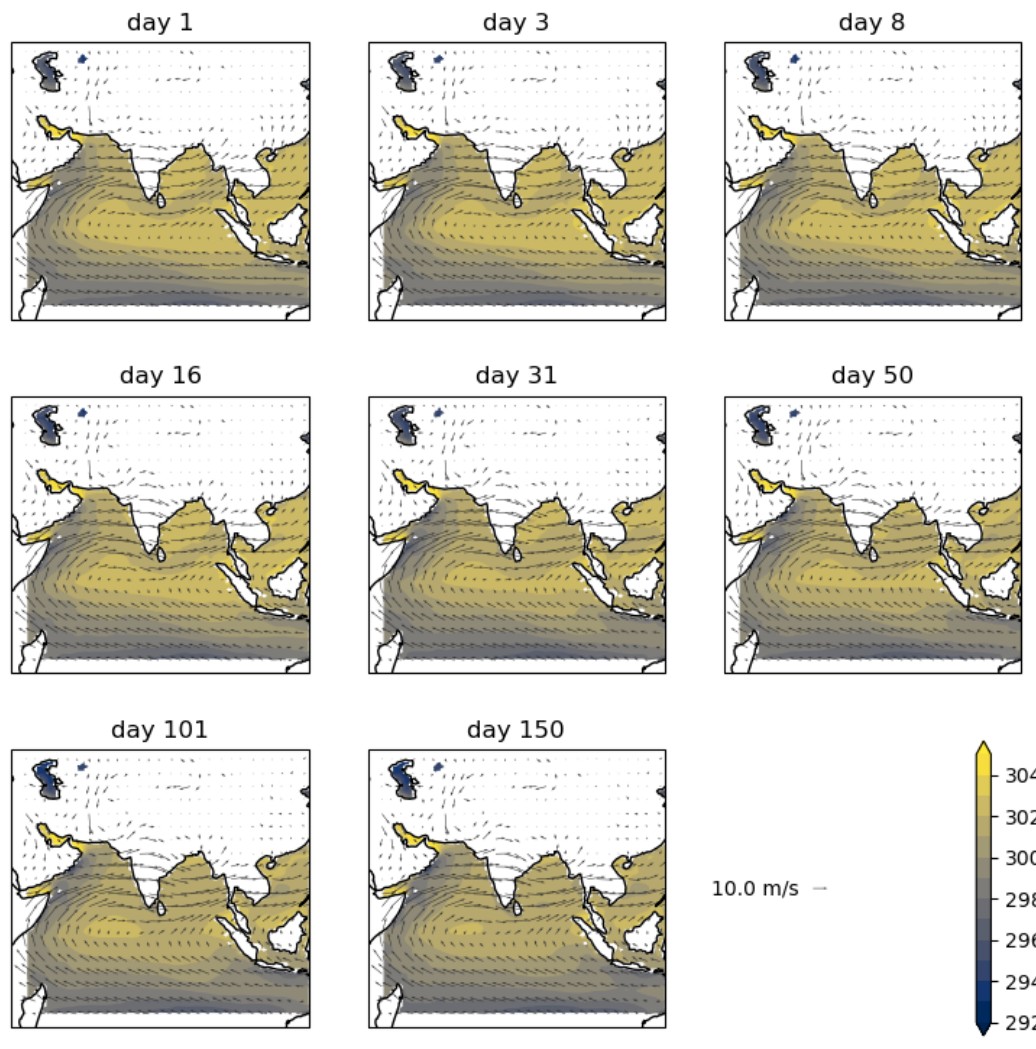

**Figure A9. SST (colours, in K) and 850-hPa wind (vectors) in GloSea5, at the hindcast lead time shown, averaged over June–August 2002–2015.**

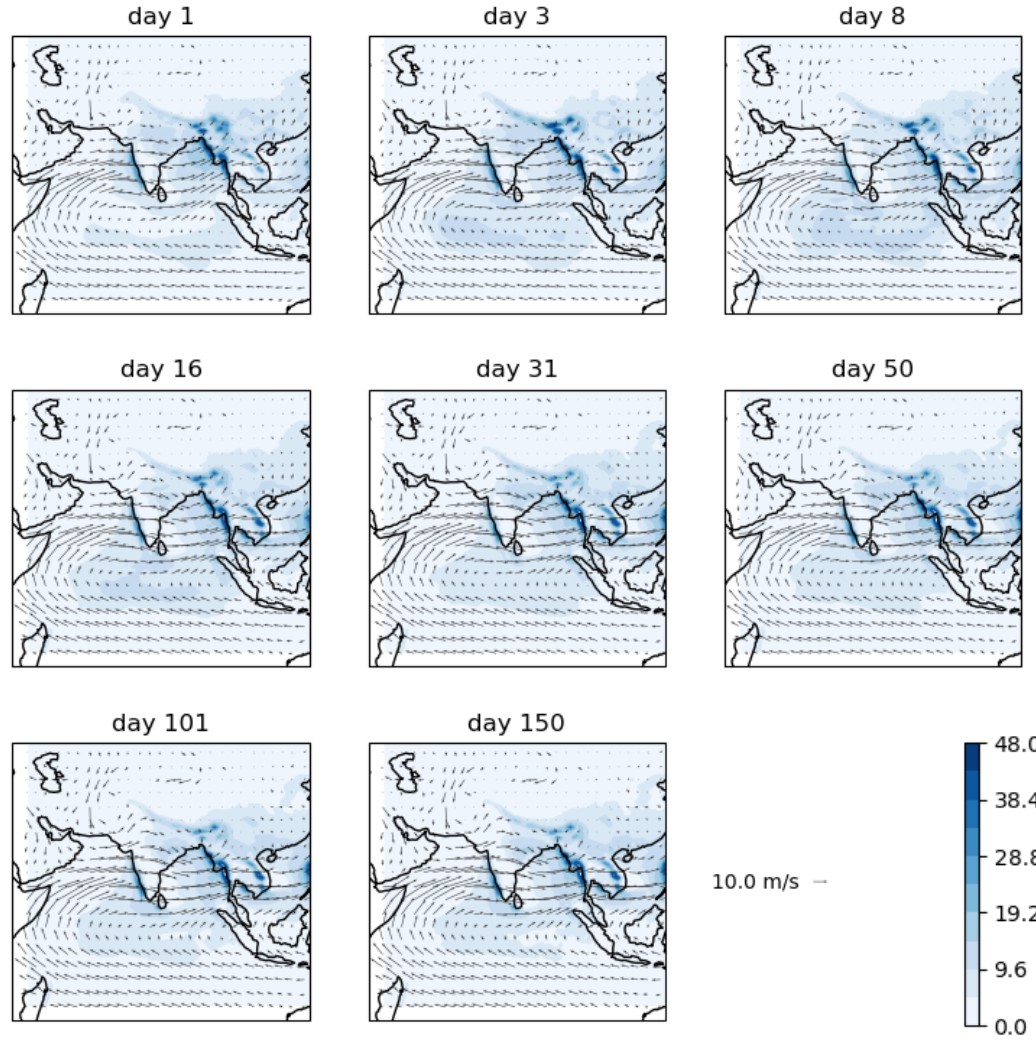

**Figure A10. Precipitation (colours, in mm/day) and 850-hPa wind (vectors) in GloSea6, at the hindcast lead time shown, averaged over June–August 2002–2015.**

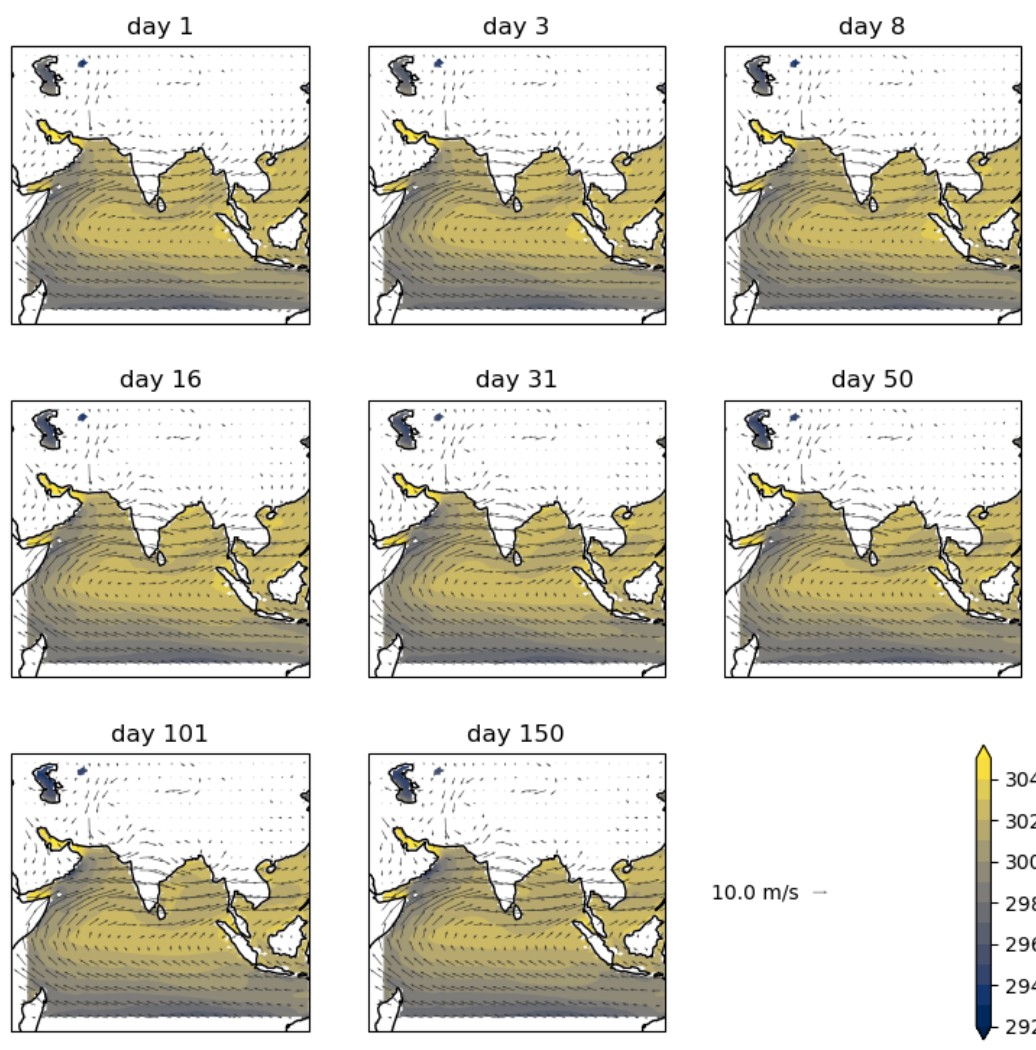

**Figure A11. SST (colours, in K) and 850-hPa wind (vectors) in GloSea6, at the hindcast lead time shown, averaged over June–August 2002–2015.**

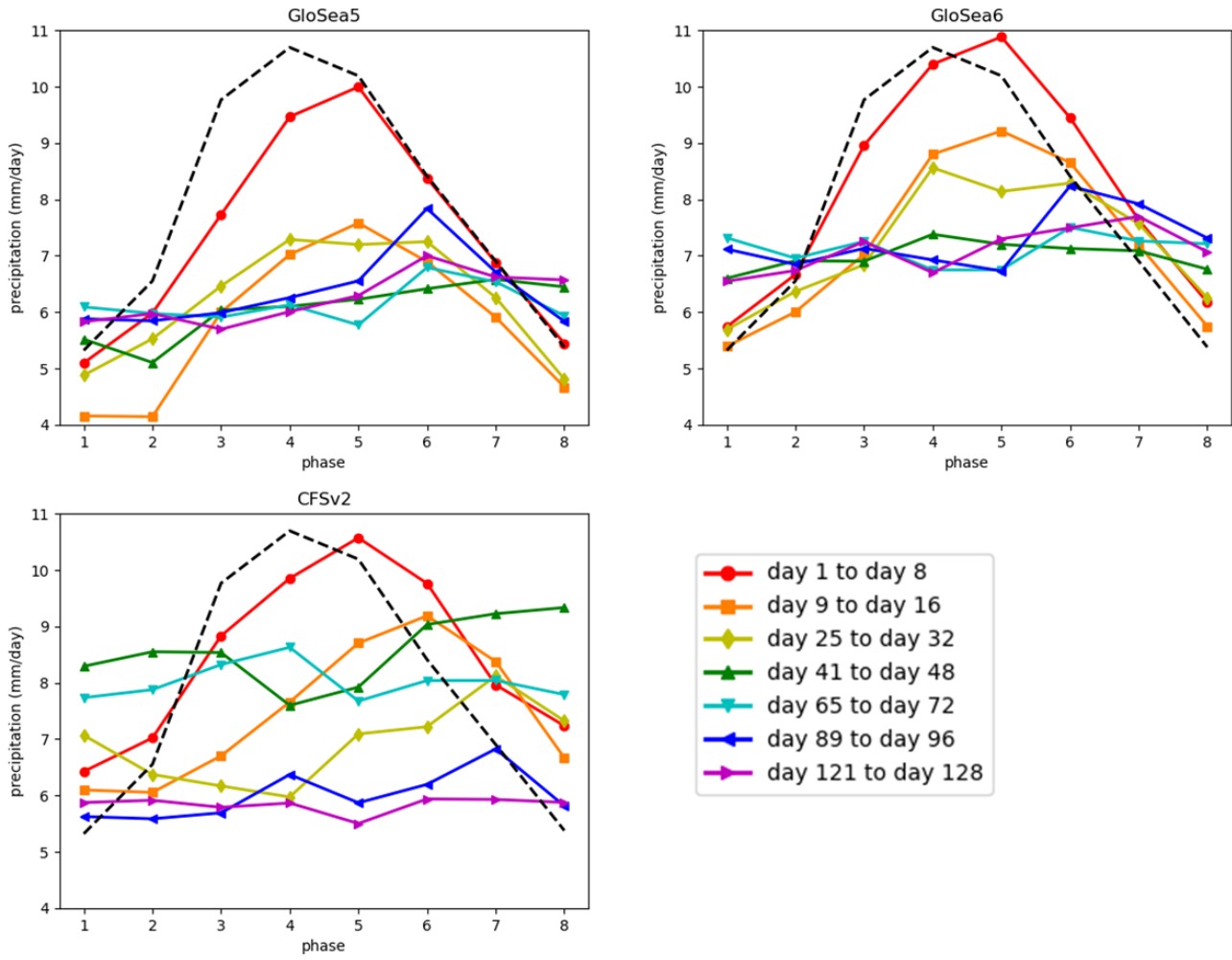


**Figure A12. As Fig. 13 but with only 3 of the 7 ensemble members in GloSea.**

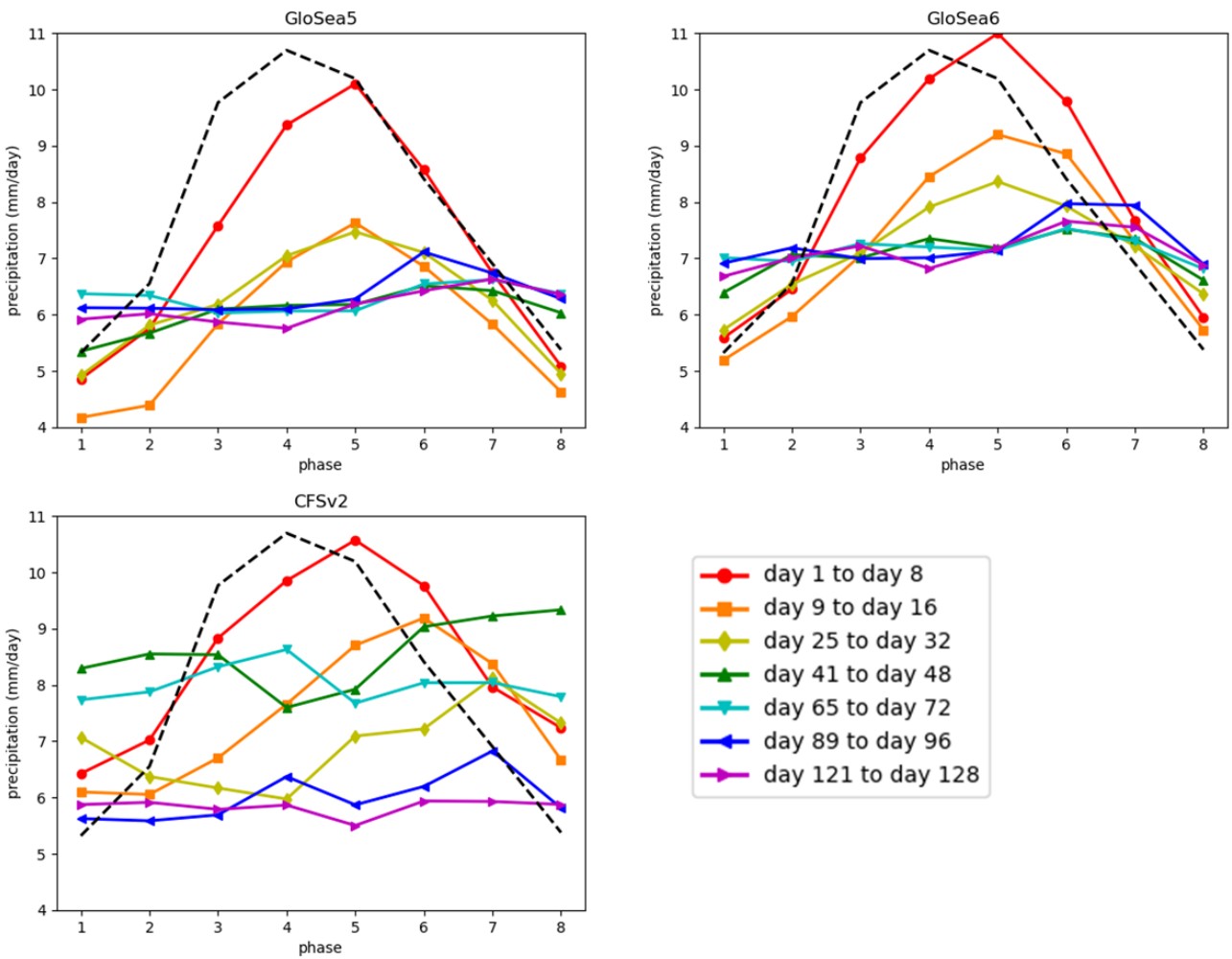

**Figure A13. As Fig. 13 but with GloSea5 hindcasts valid during 1994–2015 and GloSea6 hindcasts valid during 1994–2016 (all other data unchanged, i.e., the black dashed lines refer to observations during 2002–2015 in all three panels).**

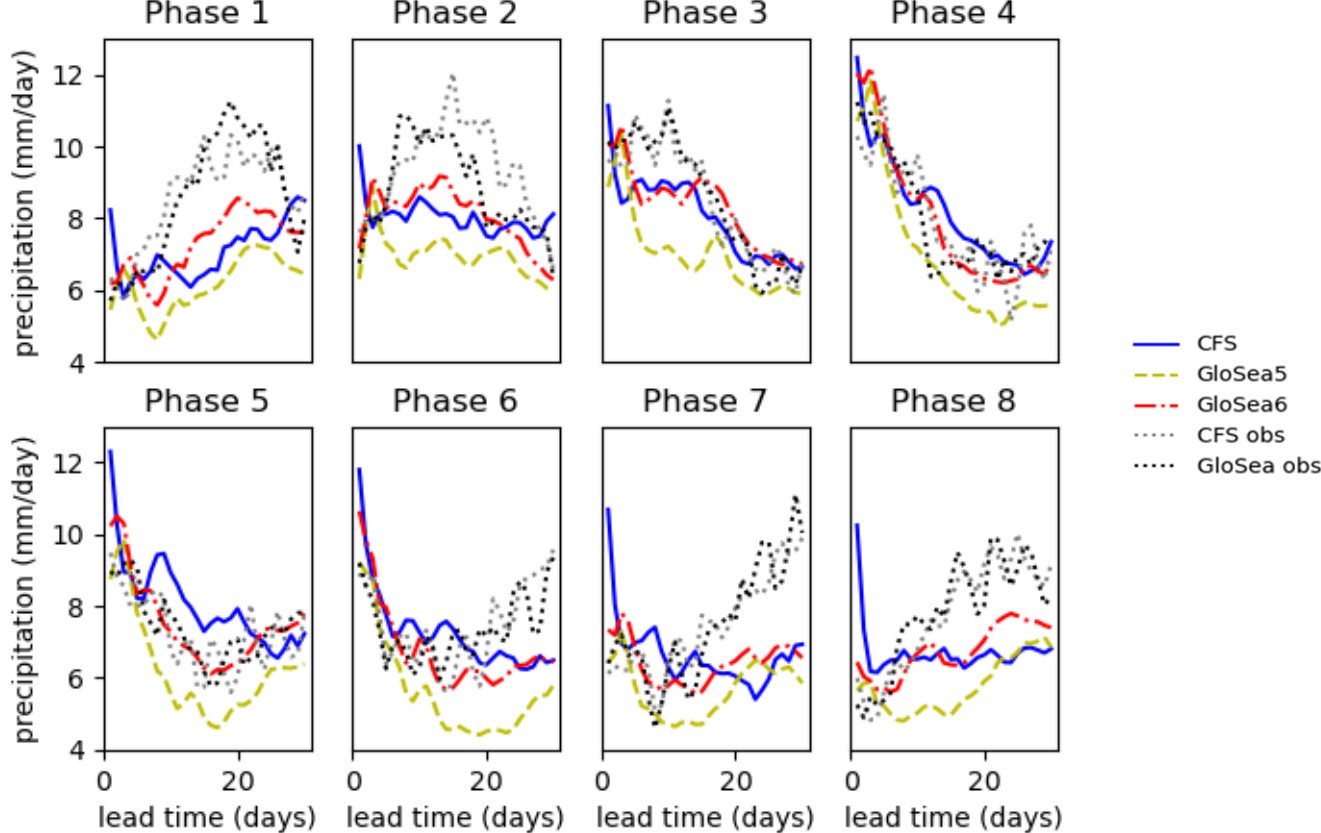

**Figure A14. As Fig. 14 but including start dates up to 31st August and only to 30 days' hindcast time.**

## Code availability

Due to intellectual property right restrictions, we cannot provide either the source code or documentation papers for the Met Office Unified Model (MetUM). The MetUM is available for use under licence. For further information on how to apply for a licence, see https://www.metoffice.gov.uk/research/approach/collaboration/unified-model/partnership. JULES is available under licence free of charge. For further information on how to gain permission to use JULES for research purposes, see https://jules.jchmr.org/. The model code for NEMO v3.4 is available from the NEMO Consortium and can be downloaded from their repository (https://forge.ipsl.jussieu.fr/nemo/chrome/site/doc/NEMO/guide/html/install.html; https://doi.org/10.5281/zenodo.1464816; NEMO System Team, 2020). The model code for CICE is freely available from the CICE Consortium, a group of stakeholders and primary developers of the Los Alamos sea ice model, and can be downloaded

from the CICE repository (https://github.com/CICE-Consortium/CICE/wiki). To obtain source code and documentation for
CFSv2, see https://www.tropmet.res.in/monsoon/monsoon2/MM_Model_CFS_Output.php. Model data used in this study are available to research collaborators upon request. Observed precipitation data were obtained from ftp://arthurhou.pps.eosdis.nasa.gov/pub/gpmdata/YYYY/MM/DD/imerg/ and observed BSISO data were taken from https://iprc.soest.hawaii.edu/users/kazuyosh/ISO_index/data/BSISO_25-90bpfil_pc.extension.txt. Observed SST data were obtained from https://cds.climate.copernicus.eu/cdsapp#!/dataset/satellite-sea-surface-temperature?tab=overview and
reanalysed wind data were obtained from https://cds.climate.copernicus.eu/cdsapp#!/dataset/reanalysis-era5-pressure-levels?tab=overview, using code provided at https://cds.climate.copernicus.eu/cdsapp#!/software/app-c3s-daily-era5-statistics?tab=overview to produce daily quantities.

**Author contribution**

RJK designed the evaluation, with contributions from all authors. AS generated and evaluated the CFSv2 hindcasts and RJK
evaluated the GloSea hindcasts. RJK prepared the manuscript, with contributions from all authors.

**Competing interests**

The authors declare that they have no conflict of interest.

**Acknowledgements**

R.J. Keane and G.M Martin were funded by the Met Office Weather and Climate Science for Service Partnership (WCSSP)
India project which is supported by the UK Department for Science, Innovation & Technology (DSIT). WCSSP India is a collaborative initiative between the Met Office and the Indian Ministry of Earth Sciences (MoES). A. Srivastava is supported by the Indian Institute of Tropical Meteorology, Ministry of Earth Sciences, Pune, India. The IMERG data were provided by the NASA/Goddard Space Flight Center and PPS, which develop and compute IMERG as a contribution to GPM, and archived at the NASA GES DISC. Plots including ERA5 and observed SST data were generated using Copernicus Climate Change
Service information [2024] and software for calculating daily statistics. Neither the European Commission nor ECMWF is responsible for any use that may be made of the Copernicus information or data it contains.
The authors would like to thank Rajib Chattopadhyay and one anonymous reviewer for valuable comments which have greatly improved the quality of the manuscript.

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
