# Peer review of "Development of Indian summer monsoon precipitation biases in two seasonal forecasting systems and their response to large-scale drivers"

_EGUsphere, 2023_

## Author Comment (AC1)

We would like to thank both reviewers for their thorough assessment of our submitted manuscript and for their recommendations for improvements. We respond here to comments from each reviewer in turn, and have produced a revised manuscript and a document highlighting the differences between the originally submitted manuscript and the revised manuscript. We refer to these two accompanying documents here.

RC1:

The manuscript compares the skills of CFSv2, Glosea5, and Glosea6 in a seasonal forecasting setup during the monsoon season. The objectives are clearly stated, and the manuscript is well written in explaining the results. The study found an initial reduction in precipitation followed by a recovery associated with an increasingly cyclonic wind field to the north-east of India in all three models. Similarly, they compared the skills of BSISO modes and the skills of the model initialized in different months. I have a few comments that would help to improve the organization of the manuscript and bring out some more clarity in the manuscript. The comments are given below:

(a) The text (P4 L96) mentioned hindcasts from 2002–2015, whereas the Fig. 1 (and some other figure) caption mentions 2012–2015. Please clarify.

Thank you for noticing this, it was a mistake! The captions have now been corrected to show 2002—2015.

(b) The spatial bias plots in Figs. 2–6 could be arranged in such a fashion that the biases are easily compared for different models. I suggest that the three models be compared together in the same figure. One column for CFSv2, one for GLosea5, and one for Glosea6, with each row showing the bias in different lead times. There are seven bias panels. Hence, if needed, multi-page panels can be designed to preserve the scientific quality of the description.

These have now been rearranged as Figs. 6—9.

(c) Beyond a month or so, the difference (bias) plot (e.g., day31 - day16) or day101-day50) can be of little use for inter-model comparison. 101st day forecast or 51st day forecast of rainfall itself has little meaning for intermodel comparison and for operational use. Probably a monthly average centered around those days would give a better idea about large-scale biases.

For the difference 50—101 days onwards, we take an average from n-15 to n+15 for each lead time n (for both start and end of "period"), as plotted in Figs. 8—9.

(d) The study does not evaluate standard metrics for seasonal forecasts. Some comparative idea of the seasonal cycle, its biases or diurnal cycle averaged at higher lead times has to be given. Also, a 3–4 month averaged mean monsoon basic state bias can be compared for clarity.

We have added a new subsection to the Results section, 3.1, where we present overall biases for different start months and the seasonal cycle for each lead time.

(e) The mm/hr unit is used here in the figures. Mostly for monsoon forecasts and operational uses, mm/day is used and gives a standard metric of evaluation to understand the subseasonal variability. It would be good to use the mm/day unit in biases to have a direct comparison with earlier papers.

We have updated the precipitation unit to mm/day.

(f) Also, any comment on the ENSO-Monsoon relationship or the IOD-Monsoon relationship? would be a good addition for model comparison.

We have added some discussion on this to the final section, lines 502—509 (D540—D547).

RC2:

Review of "Development of Indian summer monsoon precipitation biases in two seasonal forecasting systems and their response to large-scale drivers" by Keane et al.

Synopsis:

The paper by Keane et al. compares ISM biases in the Met Office Global Coupled Model (GC) and the NCEP Climate Forecast system (CFSv2). Both models develop a dry bias in the first two weeks of the forecast which is followed by a reduction of the dry bias. This "recovery" is more pronounced in CFSv2. After the recovery and at lead times beyond roughly 70 days both systems exhibit a similar dry bias. Overall the manuscript is well written and the results are mostly clear. However, given the scope of WCD, the paper is very technical in its current form and in my view lacks physical interpretation of the findings. Even if a quantitative analysis of physical processes is beyond the scope of the study, a plausible explanation of the findings or the formulation of physically justified hypotheses would certainly strengthen the paper. In view of the fact that this requires major changes to the text and potentially a deeper analysis of the material, I recommend major revisions before publication in WCD.

Major:

1) In large parts the current paper reads more like a technical report. Given that the paper is not overly lengthy yet I would like to encourage the authors to also include physical interpretations of the results. Questions that one may ask are: Is it possible to develop a process-level understanding of the interplay between SST biases over the Arabian Sea and the dry bias over India? Is is too litte moisture advection or a

lack of moisture sources related to the low SSTs? What process may be responsible for the marked recovery in CFSvs in July and August but not in June? Also, it may be helpful to refer to previous studies analysing the representation of the ISM in CFSv2 (e.g., Narapusetty et al, 2015; Hari Prasad et al. 2021). The authors highlight that GloSea6 has smaller biases compared to GloSea5. Of course this improvement may be due to several factors, but given the authors' insight into the model, is it possible to at least hypothesize which model update may help to explain the improvement?

We tried to provide some interpretation of the results in the original draft, but accept that this aspect of the manuscript should be improved, in particular with a consideration of possible physical explanations. We have added some as subsection 3.2.2, looking at the development of key quantities as a function of forecast lead time (now shown as Fig. 10), and how they vary within the model (Fig. 11) and put forward some physical interpretations of this behaviour. We have also added some analysis of how the interpretation varies for different parts (months) of the season, in Section 3.3. In terms of the reasons for the improvement in GloSea6, we have added some text in lines 444—451 to discuss the possible reasons for this.

2) This is actually more a technical comment but it may affect the interpretation of the results. The changes in SST and precipitation shown in Fig. 9 & 10 seem to be identical. By visual inspection I tried to find differences but I couldn't. It would be very important to bouble-check if the figures are correct and how this affects the interpretation of the results. Please excuse me if I am mistaken here.

Yes it looks like we accidentally included the same figures – apologies (although the text was based on looking at the correct figures so should have been correct!). We have removed this part as it is superseded by the analysis written in response to point 1.

Minor:

l. 44: Can you please specify what you mean by "shorter lead times"?

This has been made more explicit.

l. 50: Would it be possible to mention where the cold SST biases typically occur?

This has been made more explicit (the biases are shown as being fairly widespread across the northern Indian Ocean).

l. 59: This sentence could be easier to understand if written "... an intriguing finding that CFSvs produces better ISM forecasts at longer lead times than at shorter lead times".

Thank you, we have changed it to your suggested formulation.

l. 78: I assume that only data from February-August are analysed but not the entire year. It this correct?

The start times can be as early as November, but yes it is not the whole year that is analysed. Hopefully this becomes clear as the text continues, but we have adjusted the text here to try to make it clearer on first reading.

l. 148: Please consider highlighting the averaging box over India in Figs. 2-7. Also, I assume it should be 2002-2015 (cf. l. 129). If so, the same error appears also in the captions of Figs. 2-8, 11, 12.

This is now done in Figs. 6—9. Thank you for picking up the error – this has now been corrected.

l. 157: It would be helpful to the reader if the actual leadtimes were given instead of "the next five days", "the following eight days" etc.

This is now made more explicit.

l. 208: Though performing CFSv2 hindcasts at lower resolution, Hari Prasad et al. 2021 documented similar biases. Perhaps you can refer to their results.

This is now referred to in lines 237—238 (D253—D254).

l. 217: Please specify what "more" is referring to.

This just meant more air advected; hopefully it is clear now.

l. 287: I very much agree with that interpretation. As the authors themselves write in the conclusions: Would it be possible to quantify this interpretation by evaluating the prediction skill of the BSISO in the simulations? At least studies could be referenced that have already assessed the skill of the BSISO. As an alternative one could investigate the relation between predicted BSISO phases and precipitation, e.g., for a predicted BSISO phase 4 at 60 days lead time (independent on whether it is actually observed or not), would we find the same maximum precipitation value as during observed phases 4 of the BSISO?

These are interesting questions; we have added some text and references to the paragraph where they are discussed in the final section, lines 493—501 (D530—D539).

l. 361: Do I understand correctly that precipitation values in Fig. 12 are not bias corrected? How would the interpretation of the results change if a bias correction was applied?

Yes that is correct. We have added some text on lines 422—427 (D458—D463) discussing how the results might look in the bias-corrected seasonal forecasts.

Technical:

Fig. 1: The green vertical line at 50 days is missing.

Thank you, this has been added.

References:

Hari Prasad, K. B. R. R., Ramu, D. A., Rao, S. A., Hameed, S. N., Samanta, D., & Srivastava, A. (2021). Reducing systematic biases over the Indian region in CFS V2 by dynamical downscaling. Earth and Space Science, 8, e2020EA001507. https://doi.org/10.1029/2020EA001507

Jie, W., Vitart, F., Wu, T. and Liu, X. (2017), Simulations of the Asian summer monsoon in the sub-seasonal to seasonal prediction project (S2S) database. Q.J.R. Meteorol. Soc., 143: 2282-2295. https://doi.org/10.1002/qj.3085

Narapusetty, B., Murtugudde, R., Wang, H. et al. Ocean–atmosphere processes driving Indian summer monsoon biases in CFSv2 hindcasts. Clim Dyn 47, 1417–1433 (2016). https://doi.org/10.1007/s00382-015-2910-9

These are now referred to in the manuscript.

---

## Author Response (AR2)

Thanks for the thorough revision that has greatly improved the paper. I think that this is nearly ready for publication. Please address the following three points:
1) Make sure you always write GloSea with a capital "s" (see P8 for example).

Thank you, we have now corrected this.

2) Section 3.1: This section contains 4 multi-panel figures but only about 10 lines of text to describe and discuss them. This is too little to justify the figures' existence. I would suggest to either move some figures to the Appendix or to expand the text somewhat giving more details on the differences between the models.

We have moved Figures 1—3 to the appendix, but incorporated one panel from each into Figure 4, to produce a new Figure 1. We have also rearranged the subsections and subsubsections in Section 3.

3) Geographical terms: Please be consistent with capitalization. E.g. on P26 you write Tropical Indian Ocean and then equatorial Indian Ocean (which by the way you earlier defined an abbreviation for). So please be consistent and also use abbreviations consistently.

Thank you for picking this up, we have replaced "equatorial Indian Ocean" with "EIO" at that instance and the capitalisation is now consistent between "equatorial" and "tropical" (uppercase when part of a name, e.g., "EIO", and lowercase otherwise). We have also replaced some instances of "Indian summer monsoon" with "ISM" and of "[sea] surface temperature" with "SST".

We have also taken the opportunity to make some other minor adjustments to the text: hopefully these will be clear from the differences document.